Metagenomic investigation of the equine faecal microbiome reveals extensive taxonomic diversity

Gilroy Rachel 1
http://orcid.org/0000-0001-8347-0339 Leng Joy 2
Ravi Anuradha 1
http://orcid.org/0000-0003-4826-5406 Adriaenssens Evelien M. 1
Oren Aharon 3
Baker Dave 1
La Ragione Roberto M. 2
Proudman Christopher 2
http://orcid.org/0000-0003-1807-3657 Pallen Mark J. 1 2 4 mark.pallen@quadram.ac.uk
1 Quadram Institute Bioscience , Norwich , United Kingdom
2 School of Veterinary Medicine, University of Surrey , Guildford , United Kingdom
3 The Institute of Life Sciences, Hebrew University of Jerusalem , Jerusalem , Israel
4 University of East Anglia , Norwich , United Kingdom
Syed Mudasir Ahmad
Electronic publication date: 2022 Mar 23
Publication date: 2022
Volume: 10
Electronic Location ID: e13084
Received 2021 Nov 3; Accepted 2022 Feb 17
Copyright: © 2022 Gilroy et al.
Copyright year: 2022
Copyright holder: Gilroy et al.
License: This is an open access article distributed under the terms of the Creative Commons Attribution License, which permits unrestricted use, distribution, reproduction and adaptation in any medium and for any purpose provided that it is properly attributed. For attribution, the original author(s), title, publication source (PeerJ) and either DOI or URL of the article must be cited.
License URL: https://creativecommons.org/licenses/by/4.0/

Keywords: Equine, Microbiome, Metagenomics, Taxonomy, Sequencing

Funding: Biotechnology and Biological Sciences Research Council Quadram Institute Bioscience BBSRC-funded Strategic Programme, Microbes in the Food Chain BB/R012504/1 Theme 3, Microbial Communities in the Food Chain BBS/E/F/000PR10351 Medical Research Council CLIMB MR/L015080/1 BBSRC Institute Strategic Programme Gut Microbes and Health BB/R012490/1, BBS/E/F/000PR10353, BBS/E/F/000PR10356 Alborada Trust The work was supported by the Biotechnology and Biological Sciences Research Council (BBSRC). Rachel Gilroy, Anuradha Ravi and Mark J Pallen are supported by the Quadram Institute Bioscience BBSRC-funded Strategic Programme: Microbes in the Food Chain (project no. BB/R012504/1) and its constituent project BBS/E/F/000PR10351 (Theme 3, Microbial Communities in the Food Chain) and by the Medical Research Council CLIMB grant (MR/L015080/1), Evelien M Adriaenssens was funded by the BBSRC Institute Strategic Programme Gut Microbes and Health BB/R012490/1 and its constituent projects BBS/E/F/000PR10353 and BBS/E/F/000PR10356. Joy Leng and Christopher Proudman were funded by the Alborada Trust (http://www.alboradatrust.com) as part of their Alborada Well Foal study. The funders had no role in study design, data collection and analysis, decision to publish, or preparation of the manuscript.

==============================
Background

The horse plays crucial roles across the globe, including in horseracing, as a working and companion animal and as a food animal. The horse hindgut microbiome makes a key contribution in turning a high fibre diet into body mass and horsepower. However, despite its importance, the horse hindgut microbiome remains largely undefined. Here, we applied culture-independent shotgun metagenomics to thoroughbred equine faecal samples to deliver novel insights into this complex microbial community.

Results

We performed metagenomic sequencing on five equine faecal samples to construct 123 high- or medium-quality metagenome-assembled genomes from Bacteria and Archaea. In addition, we recovered nearly 200 bacteriophage genomes. We document surprising taxonomic diversity, encompassing dozens of novel or unnamed bacterial genera and species, to which we have assigned new Candidatus names. Many of these genera are conserved across a range of mammalian gut microbiomes.

Conclusions

Our metagenomic analyses provide new insights into the bacterial, archaeal and bacteriophage components of the horse gut microbiome. The resulting datasets provide a key resource for future high-resolution taxonomic and functional studies on the equine gut microbiome.

Introduction

The horse has played a crucial role in human development and in the extension of human settlement (Roberts, 2017). Domestication of the horse began at least 6,000 years ago and led to diversification into numerous breeds, accompanied by significant biological changes (Fages et al., 2019). The horse remains an important component of human society, with around 60 million horses worldwide (Clarkson, 2017). Horses provide health benefits through horse-riding and equine-assisted therapy alongside playing roles as working animals across the globe, in transport, agriculture or policing. The horse remains an important food animal globally, with five million animals slaughtered for food each year and horsemeat now in favor as a low-methane red-meat alternative to beef (Belaunzaran et al., 2015). In the UK, there are around 374,000 horse-owning households and horseracing is the second most attended sport in the country after football, contributing £4.7 billion to the UK economy (British Equine Trade Association, 2019).

As a foraging herbivore, the horse relies on a cellulose-rich diet of grass and legumes. However, unlike cattle, horses have no rumen to digest complex carbohydrates. Instead, they rely on hindgut fermentation: an efficient but enigmatic process—far less well understood than ruminal digestion—that relies on a rich microbial community, the hindgut microbiome, encompassing bacteria, archaea and viruses, together with fungi and other eukaryotic microbes (Costa & Weese, 2018; Julliand & Grimm, 2016; Santos et al., 2011). This ecosystem plays a key role in nutrient assimilation and feed conversion—effectively turning grass into horseflesh and horsepower. The horse gut also acts as a reservoir of equine and several human pathogens, as well as sources of antimicrobial resistance (Maddox et al., 2015).

Crucially, various diseases are associated with disturbances in hindgut microbial ecology, including foal diarrhoea, colitis, laminitis, colic and equine grass sickness (Leng et al., 2018). Thus, by better understanding the equine hindgut microbiome, we stand to inform interventions that can improve the health and welfare, performance, value and longevity of horses.

Previous studies of the horse hindgut microbiome have documented a rich variety of microorganisms (spanning phyla from all three domains of life) and have shown that the taxonomic composition of this community varies with age, breed and disease status and has changed during domestication (Costa & Weese, 2018; Julliand & Grimm, 2016; Leng et al., 2018; Massacci et al., 2020; O’Donnell et al., 2013; Proudman et al., 2015; Stewart et al., 2018; Edwards et al., 2020; Metcalf et al., 2017; Leng et al., 2019). However, earlier studies have largely relied on short-read meta-barcoding analyses of 16S rRNA gene sequences, which are limited in that they fail to provide resolution down to the species or strain level, provide limited insight into population structures or functional repertoires of microbial species and fail to cover viruses and eukaryotes. Thus, despite previous efforts—and drawing on comparisons with the human microbiome, where new species are still being discovered (Almeida et al., 2019; Forster et al., 2019)—the horse hindgut microbiome presents us with a vast, only superficially explored (Di Pietro et al., 2021) landscape of taxonomic, ecological and functional diversity, certain to encompass important, yet undiscovered roles. Babenko et al. (2020) emphasize this with their preliminary exploration of the equine faecal virome, presenting a rich taxonomically diverse viral community which is thought to be essential in shaping microbial ecology. As in studies of the human gut microbiome, faeces provides ready non-invasive access to the gut contents. Application of short-read metagenomics to complex environmental microbial communities has proven capable of recovering large-scale catalogues of near-complete genomes, vastly expanding the tree of life to include multiple phyla with no known cultured representative (Parks et al., 2017). Drawing on these principles, as a component within the Alborada Well Foal study—a cohort study of equine gut microbial development and health—we applied shotgun metagenomics to five equine faecal samples from 12-month-old thoroughbreds to expand our knowledge of this microbial landscape.

Materials and Methods

Sample collection and storage

Faecal samples were from five, 12-month-old Thoroughbred racehorses from the same farm and field in Ireland. All samples were collected in April 2019 from horses raised on permanent pasture of mixed ryegrass. Horses were not being exercised at the time of sample collection. Feed supplementation whilst at pasture was proprietary post weaning cereal and trace element pellets plus an additional trace mineral and amino acid supplement. All horses had received ivermectin and praziquantel paste four weeks prior to sampling. Samples were collected as part of the Alborada Well Foal study, under the University of Surrey’s ethical review framework, project code: NERA-2017-007-SVM. 100 g of freshly evacuated faeces was collected from each horse in sterile tubes before immediate storage at 4 °C on site at the stud. All samples were shipped the same day at ambient temperature and received within 24 h. Upon receipt, samples were refrigerated before being aliquoted and stored at −80 °C until DNA extraction. Samples were thawed and homogenized before DNA extraction using the DNeasy PowerSoil kit (Qiagen), following manufacturer’s instructions. Extracted DNA was stored at −20 °C before further analysis.

Metagenomic sequencing and processing

Illumina sequencing libraries were constructed as previously described by Ravi et al. (2019). Paired-end metagenomic sequencing was performed on the Illumina NextSeq, before bioinformatic processing on the Cloud Infrastructure for Microbial Bioinformatics (CLIMB) (Connor et al., 2016). Output reads (2 × 150 bp) were assessed for quality using FastQC v0.11.8 and then trimmed using Trimmomatic v0.36 configured to a minimum read length of 40 (Andrews, 2019; Bolger, Lohse & Usadel, 2014). All metagenomic samples described here can be accessed on the Sequence Read Archive under BioProject ID PRJNA590977. Reads were aligned to the horse genome (GCF_002863925.1) using Bowtie2 v2.3.4.1 (Langmead & Salzberg, 2012), allowing removal of host reads with SAMtools v1.3.1 (Li et al., 2009).

Taxonomic profiling of sequencing reads was performed using Kraken 2 (Wood, Lu & Langmead, 2019) to search a microbial database built from archaeal, bacterial, fungal, protozoan, viral and univec_core sequences in Refseq in January 2021. Bracken was used to estimate taxon abundance from Kraken 2 profiles, accepting only those taxa with >1,000 assigned reads (Lu et al., 2017). Bracken-database files were generated using “bracken-build” on our microbial database and visualised using Pavian (Breitwieser & Salzberg, 2016).

Metagenomic assembly and binning

Host-depleted reads were assembled individually from each metagenomic sample with MegaHIT (Li et al., 2016), using kmer sizes 25, 43, 67, 87 & 101, before assessing the quality of resulting contiguous sequences (contigs) with anvi’o v7 (Eren et al., 2015). Filtered reads from each sample were mapped against the associated assembly to provide an estimate of contig abundance using Bowtie 2 (Langmead & Salzberg, 2012). Resulting Sequence alignment/map (SAM) files were converted to binary alignment/map (BAM) files before being sorted and indexed using SAMtools (Li et al., 2009). Contig coverage depth was translated from each BAM file, before separately binning contigs >1,000 bp with MaxBin v2.2.6 (Wu, Simmons & Singer, 2016) and CONCOCT v1.1.0 (Alneberg et al., 2014) and binning contigs >1,500 bp with MetaBAT 2 v2.12.1 (Kang et al., 2019).

DAS Tool was applied to the output from all three bin predictors, generating a catalogue of 196 bins from five samples (Sieber et al., 2018). All bins were profiled against the BAM file for their source metagenomic sample using the anvi’o ‘anvi-profile’ workflow (Eren et al., 2015). Using the ‘anvi-interactive’ tool, each bin was refined manually according to GC content, single copy core gene (SCG) taxonomy and coverage as well as detection statistics. CheckM v1.0.11 (Parks, Imelfort & Skennerton, 2015) was used for quality assessment of all bins using the lineage_wf function. Bins showing >50% completion and <10% contamination were assessed for quality score (defined as estimated genome completeness score minus five times estimated contamination score), a commonly used standard for defining acceptable bin quality (Parks et al., 2017). Bins with <70% completion and/or a quality score of <50 were categorised as low-quality metagenome-assembled genomes (MAGs) (n = 29); those with >70% completion, <10% contamination and quality score >50 were categorised as medium-quality MAGs (n = 68) and those with >90% completion, <5% contamination and quality score >50 were classified as high-quality MAGs (n = 55).

Taxonomic and phylogenetic profiling of MAGs

Medium- and high-quality MAGs from all five samples were de-replicated at 95% average nucleotide identity (ANI) with a default aligned fraction of >10% using dRep v2.0 (Olm et al., 2017), to create a non-redundant species catalogue. Clustering at 99% ANI was used to identify a non-redundant strain catalogue and select a representative MAG per strain. CompareM v0.1.1 (Oksanen et al., 2019) was used to assign Average Amino-acid Identity (AAI) values followed by AAI clustering at 60% to allow delineation at the genus level.

The Genome Taxonomy Database Toolkit (GTDB-Tk) v1.5.0 (Chaumeil et al., 2019), the Contig Annotation Tool (CAT/BAT) v5.2.3 (von Meijenfeldt et al., 2019) and ReferenceSeeker v1.4 (Schwengers et al., 2020) were used to perform taxonomic assignment of representative MAGs at strain-level compared to the ‘GTDB release 202’, ‘NCBI nr (2021-01-07)’ and ‘NCBI RefSeq release 201’ databases, respectively. Where taxonomic assignments differed between GTDB-Tk, CAT/BAT or ReferenceSeeker, GTDB-Tk assignments took precedence. Only when no species-level GTDB taxonomy was available did we adopt assignments according to CAT/BAT or ReferenceSeeker (6% of assignments). Phylogeny for our final de-replicated catalogue of MAGs was performed by aligning and concatenating a set of sixteen ribosomal protein sequences (ribosomal proteins L1, L2, L3, L4, L5, L6, L14, L16, L18, L22, L24, S3, S8, S10, S17 and S19), an approach previously used to reconstruct the tree of life (Hug et al., 2016). Ribosomal sequences were extracted using anvi’o before alignment using MUSCLE v3.8.155 (Edgar, 2004) and refinement using trimAl v1.4 (Capella-Gutiérrez, Silla-Martínez & Gabaldón, 2009). A maximum-likelihood tree was constructed using FastTree v2.1 (Price, Dehal & Arkin, 2010). All novel MAG species clusters were confirmed as monophyletic, drawing on all publicly available genomes from the genus to which they had been assigned by GTDB (with genomes retrieved from NCBI). Proteomes were predicted using Prodigal v2.6.1 (Hyatt et al., 2010) before comparison against 400 universal marker proteins using PhyloPhlAn v3.0.58 (Asnicar et al., 2020) in accordance with diamond v0.9.34 (Buchfink, Xie & Huson, 2015). Multiple sequence alignment and subsequent refinement was performed using MAFFT v7.271 (Katoh et al., 2002) and trimAl v1.4 (Capella-Gutiérrez, Silla-Martínez & Gabaldón, 2009; Stamatakis, 2014). All trees were subsequently visualised and manually annotated using iTol v5.7.

Abundance and metabolic profiling of MAGs

To estimate the proportion of reads within each BioSample represented by our final, de-replicated MAG catalogue, contigs from the non-redundant MAG catalogue were concatenated and filtered reads aligned back to this MAG database using Bowtie 2 (Langmead & Salzberg, 2012). Ordered BAM files were assessed using anvi’o (Parks, Imelfort & Skennerton, 2015) to calculate coverage statistics per-contig, allowing the calculation of mean coverage across each assembled genome according to methods available at: https://merenlab.org/data/2017_Delmont_et_al_HBDs/ and described by Delmont et al. (2018). Species accumulation and distribution analyses were conducted using the Vegan package in R (Oksanen et al., 2019) before visualisation using ggplot2 (Wickham, 2016).

Functional profiling of high- and medium-quality MAGs (n = 123) was performed using DRAM (Distilled and Refined Annotation of Metabolism) at a minimum contig length of 1,000 bp (Shaffer et al., 2020). Predicted amino-acid sequences identified by Prodigal in metagenome mode (Hyatt et al., 2010) were searched against KOfam, Pfam, and CAZy databases. tRNA and rRNA sequences were identified in MAGs using tRNAscan-SE (Chan & Lowe, 2019) and Barrnap v0.9, (Seemann, 2018) respectively.

Bacteriophage identification and characterisation

VirSorter v1.0.5 (Roux et al., 2015) was applied to all contigs >5 kb within each BioSample. Contig sequences classified by VirSorter as Category 1 (“most confident”) or Category 2 (“likely”) were considered for further analysis. Candidate bacteriophage sequences were assessed for completeness and contamination, using CheckV v0.7.0 (Roux, Páez-Espino & Chen, 2021), retaining only the sequences classified as “High-quality” (>90% completeness) or “complete”. These sequences were collated and de-replicated using rapid genome pairwise clustering at 95% ANI with an aligned fraction of ≥70% to generate a catalogue of bacteriophage genome sequences. For dereplication clustering, all-vs-all genome comparisons were performed using BLASTn before ANI based clustering using the ‘anicalc’ and ‘aniclust’ CheckV scripts sequentially.

Bacteriophage contigs from the catalogue were used as queries in a BLASTn search against the NCBI non-redundant nucleotide database (conducted on 21/12/2020) using an e-value of ≤1e−5. Only matches with a query cover >50% and percentage ID >70% were selected as being significant. Initial taxonomic classification of phage genomes at order and family level was performed using https://github.com/feargalr/Demovir against a viral subset of non-redundant TrEMBL database with an e-value of ≤1e−5. For each viral contig, individual coding sequences were predicted using Prodigal (Hyatt et al., 2010), before concatenation for input into vCONTACT2 v0.9.19 (Bin Jang et al., 2019) for construction of a gene-sharing network incorporating a de-replicated RefSeq database of reference prokaryotic virus genomes. The resulting network was visualised using Cystoscape v3.8.0 (Shannon et al., 2003).

Results

Reference-based profiling documents microbial diversity

Whole genome sequencing of five faecal samples derived from 12-month-old Thoroughbred horses, each yielded >6 ng/µl DNA and collectively generated >280 million paired reads or >84 Gbp of sequence data. Reads derived from the horse genome accounted for <1% of reads from each sample (Table S1). We initially analysed reads using the k-mer-based program Kraken 2, followed by refined phylogenetic analysis via the allied program Bracken. Such analyses revealed extensive novelty and diversity in the equine faecal microbiome, with >59% of sequence reads in each sample classified by Kraken as “unassigned”, i.e. from unknown organisms. Assignable reads represented all three domains of life, as well as viruses, although bacteria predominated, accounting for >89% of assigned reads in any sample (Table S2).

Bacterial reads were predominantly assigned to the four phyla in the NCBI taxonomy most associated with animal gut microbiomes—Proteobacteria, Firmicutes, Bacteroidetes and Actinobacteria. However, the Kraken 2 profiles also provided evidence of over thirty additional bacterial phyla in this ecosystem. Many of these appear to be novel in the context of the horse gut, including Deinococcus-Thermus, Thermotogae and the Candidatus phylum Cloacimonetes (also called WWE1), which has been reported almost exclusively from anaerobic fermenters and the aqueous environment (Calusinska et al., 2018; Limam et al., 2014). However, as this phylum has recently been detected in soil fertilised with manure from dairy cattle, chickens and swine and has been implicated in anaerobic digestion of cellulose, it may play important similar roles in the vertebrate gut (Limam et al., 2014; Laconi et al., 2021). Reads assigned to eukaryotes provided evidence of budding yeasts and apicoplexan parasites in these samples.

Remarkably, two samples showed a very high relative abundance of reads assigned to the genus Acinetobacter (44% and 66% of classified reads), mirroring similar findings on two healthy horses in a previous study using 16S rRNA gene sequences (Costa et al., 2012). Bracken assigns these reads to an implausible sixty-two species of Acinetobacter, which is more likely to represent misassignment of reads rather than genuine diversity within this genus in this context.

Over a hundred newly named bacterial species

We generated almost 200 non-redundant bins from single-sample assemblies using three different approaches to binning. 123 bins represent medium- or high-quality metagenome-assembled genomes (MAGs), 96 with ≥15 amino acid tRNAs (Tables S3 and S4). Genome sizes ranged from ~0.5 to 3.8 Mbp, while GC content ranged from 31% to 60%. De-replication at 95% ANI clustered MAGs into 110 species clusters, spanning ten phyla (Fig. 1A). An average of 18% of the initial, host-depleted metagenomic reads per sample were represented within the final, dereplicated MAG catalogue. According to GTDB, around half (48%) of the MAG species clusters belonged to the Bacteroidota, while just over a third (35%) belonged to the Firmicutes (split by GTDB into Firmicutes, Firmicutes_A and Firmicutes_C). Only fourteen of the bacterial species from the horse gut had been previously defined and delineated: nine with validly published Latin binomials and five simply with alphanumerical designations assigned by GTDB (these are placeholder names assigned when no well-formed Latin name exists for the species) (Table S5).

Figure 1 Taxonomic classification of 110 MAG species clusters derived from five metagenomic equine faecal samples.

(A) Depicted as a phylogenetic tree—where phylum, as assigned by GTDB, is indicated by colour range. All GTDB-tk assigned subdivisions of the Firmicutes phylum have been collapsed to a single ‘Firmicutes’ designation. The tree was based upon an alignment of 16 concatenated ribosomal proteins and constructed using FastTree. The final tree was visualised and manually annotated using the online iTOLv5.7 tool. Phylum-level taxonomy is described by branch colour according to GTDB designation (Phyla with an alphabetical suffix have been collapsed). The presence (filled) or absence (hollow) of genes associated with catalysing carbohydrate degradation (blue) or aiding in the metabolism of short chain fatty acids (red) are reported in the associated binary plot. (B) Average Nucleotide Identity (ANI) between recovered MAGs and their closest representative within the GTDB database (release 202). Only MAGs placed within a previously recognised genus, and whereby this taxonomic assignment was inclusive of an ANI measurement, are shown. Individual plots are coloured according to GTDB designated phylum, with phyla assigned an alphabetical suffix being collapsed. A dotted line is placed at 95% ANI, representing the utilised species-level boundary.

Two of the species with validly published names, Ligilactobacillus hayakitensis (synonym Lactobacillus hayakitensis) (Morita et al., 2007) and Limosilactobacillus equigenerosi (synonym Lactobacillus equigenerosi) (Endo et al., 2008), have been previously cultured from the faeces of thoroughbred racehorses and are thought to be positively associated with equine intestinal health (Morita et al., 2009). Similarly, the species Streptococcus equinus was named in the early twentieth century after its association with horse dung and has been repeatedly isolated from this source (Andrewes & Horder, 1906; Smith & Shattock, 1962). Another of the named species found among our MAGs, Treponema succinifaciens, has been reported from the equine gut by 16S studies (Daly et al., 2001), but ours represents the first report of a genome from this species in this setting.

The recently named species Acinetobacter lanii (Zhu et al., 2021) has been isolated from the Tibetan wild ass Equus kiang, but our MAG represents the first report of an association between this species and the domesticated horse. Although the genus Phascolarctobacterium is known to inhabit the horse gut (Metcalf et al., 2017; Edwards et al., 2020), here we provide the first evidence of a specific link between the horse and the species P. succinatutens, previously found in human and pig faeces (Watanabe, Nagai & Morotomi, 2012). Our MAG catalogue provides the first report in the horse of the species Pseudomonas lundensis, first isolated from meat, but now recognised as an emerging pathogen of humans (Molin, Ternström & Ursing, 1986; Scales et al., 2018).

Among our MAG species clusters, ninety-six represent new candidate species within sixty-one bacterial genera previously delineated by GTDB. The majority of these novel species had <85% ANI to their closest known representative within GTDB databases (Fig. 1B). Sixty of these genera occur in the gut microbiota of at least one additional mammalian host species. Eleven of our species that could be assigned only to the level of family fell into ten clusters (delineated at 60% AAI) representing novel candidate genera from seven different families (Table S6). The archaeal genus Methanocorpusculum is thought to play a role in methane production in the equine gut (Murru et al., 2018). Here, we have delineated a novel species from this ecosystem: Candidatus Methanocorpusculum equi.

Building on our recent efforts with the chicken gut microbiome (Gilroy et al., 2021) and with the automated creation of well-formed Latin names, we have created Candidatus names (abbreviated as Ca.) for all the unnamed taxa revealed by our metagenomic analyses (Table 1). We also created Latin names for species and genera recognised by GTDB, but previously assigned only alphanumeric designations. For taxa found only in the horse, we created names that incorporated Greek or Latin roots for this host (e.g., Ca. Equimonas). However, if searches of the GTDB and NCBI databases suggested that genera had representatives in other gut microbiomes, we opted for names that specified gut or faeces as habitat (e.g., Ca. Limimonas).

Table 1 Protologues for newly named Candidatus genera and species.

Protologues for new Candidatus taxa identified by analysis of metagenome-assembled genomes from equine faeces.	
Description of Candidatus Alistipes equi sp. nov.	
Candidatus Alistipes equi (e’qui. L. gen. masc. n. equi, of a horse)	
A bacterial species identified by metagenomic analyses. This species includes all bacteria with genomes that show ≥95% average nucleotide identity (ANI) to the type genome for the species to which we have assigned the MAG ID E3_MB2_80 and which is available via NCBI BioSample SAMN18472495. The GC content of the type genome is 40.8% and the genome length is 2.08 Mbp.	
Description of Candidatus Apopatocola gen. nov.	
Candidatus Apopatocola (A.po.pa.to’cola. Gr. masc. n. apopatos, dung; N.L. masc./fem. suffix –cola, an inhabitant; N.L. fem. n. Apopatocola a microbe associated with faeces)	
A bacterial genus identified by metagenomic analyses. The genus includes all bacteria with genomes that show ≥60% average amino acid identity (AAI) to the genome of the type strain from the type species Candidatus Apopatocola equi. This is a new name for the GTDB alphanumeric genus UBA738, which is found in diverse mammalian guts. This genus has been assigned by GTDB-Tk v1.3.0 working on GTDB Release 06-RS202 (Olm et al., 2017; Scales et al., 2018) to the order Oscillospirales and to the family Oscillospiraceae.	
Description of Candidatus Apopatocola equi sp. nov.	
Candidatus Apopatocola equi (e’qui. L. gen. masc. n. equi, of a horse)	
A bacterial species identified by metagenomic analyses. This species includes all bacteria with genomes that show ≥95% average nucleotide identity (ANI) to the type genome for the species to which we have assigned the MAG ID E1_MB2_75 and which is available via NCBI BioSample SAMN18472466. The GC content of the type genome is 59.6% and the genome length is 1.56 Mbp.	
Description of Candidatus Apopatosoma gen. nov.	
Candidatus Apopatosoma (A.po.pa.to.so’ma. Gr. masc. n. apopatos, dung; Gr. neut. n. soma, a body; N.L. neut. n. Apopatosoma, a microbe associated with faeces)	
A bacterial genus identified by metagenomic analyses. The genus includes all bacteria with genomes that show ≥60% average amino acid identity (AAI) to the genome of the type strain from the type species Candidatus Apopatosoma equi. This is a new name for the GTDB alphanumeric genus CAG-724, which is found in diverse mammalian guts. This genus has been assigned by GTDB-Tk v1.3.0 working on GTDB Release 06-RS202 (Olm et al., 2017; Scales et al., 2018) to the order Oscillospirales and to the family CAG-272.	
Description of Candidatus Apopatosoma intestinale sp. nov.	
Candidatus Apopatosoma intestinale (in.tes.ti.na’le. N.L. neut. adj. intestinale, pertaining to the intestines)	
A bacterial species identified by metagenomic analyses. This species includes all bacteria with genomes that show ≥95% average nucleotide identity (ANI) to the type genome for the species to which we have assigned the MAG ID E5_133 and which is available via NCBI BioSample SAMN18472535. This is a new name for the alphanumeric GTDB species sp003524145, which is found in diverse mammalian guts. The GC content of the type genome is 53.8% and the genome length is 1.55 Mbp.	
Description of Candidatus Apopatousia gen. nov.	
Candidatus Apopatousia (A.po.pat.ou’s.ia. Gr. masc. n. apopatos, dung; Gr. fem. n. ousia, an essence; N.L. fem. n. Apopatousia, a microbe associated with faeces)	
A bacterial genus identified by metagenomic analyses. The genus includes all bacteria with genomes that show ≥60% average amino acid identity (AAI) to the genome of the type strain from the type species Candidatus Apopatousia equi. This is a new name for the GTDB alphanumeric genus UBA9845, which is found in diverse mammalian guts. This genus has been assigned by GTDB-Tk v1.3.0 working on GTDB Release 06-RS202 (Olm et al., 2017; Scales et al., 2018) to the order Christensenellales and to the family UBA1242.	
Description of Candidatus Apopatousia equi sp. nov.	
Candidatus Apopatousia equi (e’qui. L. gen. masc. n. equi, of a horse)	
A bacterial species identified by metagenomic analyses. This species includes all bacteria with genomes that show ≥95% average nucleotide identity (ANI) to the type genome for the species to which we have assigned the MAG ID E5_MB2_6 and which is available via NCBI BioSample SAMN18472550. The GC content of the type genome is 31.9% and the genome length is 0.57 Mbp.	
Description of Candidatus Blautia equi sp. nov.	
Candidatus Blautia equi (e’qui. L. gen. masc. n. equi, of a horse)	
A bacterial species identified by metagenomic analyses. This species includes all bacteria with genomes that show ≥95% average nucleotide identity (ANI) to the type genome for the species to which we have assigned the MAG ID E4_MB2_89 and which is available via NCBI BioSample SAMN18472531. GTDB has assigned this species to a genus marked with an alphabetical suffix. However, as this genus designation cannot be incorporated into a well-formed binomial, in naming. this species, we have used the current validly published name for the genus. The GC content of the type genome is 48% and the genome length is 2.14 Mbp.	
Description of Candidatus Caballimonas gen. nov.	
Candidatus Caballimonas (Ca.bal.li.mo’nas. L. masc. n. caballus, a horse; L. fem. n. monas, a monad; N.L. fem. n. Caballimonas, a microbe associated with horses)	
A bacterial genus identified by metagenomic analyses. The genus includes all bacteria with genomes that show ≥60% average amino acid identity (AAI) to the genome of the type strain from the type species Candidatus Caballimonas caccae. This genus has been assigned by GTDB-Tk v1.3.0 working on GTDB Release 06-RS202 (Olm et al., 2017; Scales et al., 2018) to the order Christensenellales and to the family Borkfalkiaceae.	
Description of Candidatus Caballimonas caccae sp. nov.	
Candidatus Caballimonas caccae (cac’cae. Gr. fem. n. kakke, faeces; N.L. gen. n. caccae, of faeces)	
A bacterial species identified by metagenomic analyses. This species includes all bacteria with genomes that show ≥95% average nucleotide identity (ANI) to the type genome for the species to which we have assigned the MAG ID E3_31 and which is available via NCBI BioSample SAMN18472486. The GC content of the type genome is 34.9% and the genome length is 0.91 Mbp.	
Description of Candidatus Cacconaster gen. nov.	
Candidatus Cacconaster (Cac.co.nas’ter. Gr. fem. n. kakke, dung; Gr. masc. n. naster, an inhabitant; N.L. masc. n. Cacconaster, a microbe associated with faeces)	
A bacterial genus identified by metagenomic analyses. The genus includes all bacteria with genomes that show ≥60% average amino acid identity (AAI) to the genome of the type strain from the type species Candidatus Cacconaster caballi. This is a new name for the GTDB alphanumeric genus Bact-11, which is found in diverse mammalian guts. This genus has been assigned by GTDB-Tk v1.3.0 working on GTDB Release 06-RS202 (Olm et al., 2017; Scales et al., 2018) to the order Bacteroidales and to the family UBA932.	
Description of Candidatus Cacconaster caballi sp. nov.	
Candidatus Cacconaster caballi (ca.bal’li. L. gen. masc. n. caballi, of a horse)	
A bacterial species identified by metagenomic analyses. This species includes all bacteria with genomes that show ≥95% average nucleotide identity (ANI) to the type genome for the species to which we have assigned the MAG ID E2_MB2_69 and which is available via NCBI BioSample SAMN18472478. The GC content of the type genome is 50.7% and the genome length is 1.38 Mbp.	
Description of Candidatus Cacconaster equi sp. nov.	
Candidatus Cacconaster equi (e’qui. L. gen. masc. n. equi, of a horse)	
A bacterial species identified by metagenomic analyses. This species includes all bacteria with genomes that show ≥95% average nucleotide identity (ANI) to the type genome for the species to which we have assigned the MAG ID E1_MB2_89 and which is available via NCBI BioSample SAMN18472469. The GC content of the type genome is 48.5% and the genome length is 1.65 Mbp.	
Description of Candidatus Cacconaster equifaecalis sp. nov.	
Candidatus Cacconaster equifaecalis (e.qui.fae.ca’lis. L. masc. n. equus, a horse; N.L. masc. adj. faecalis, faecal; N.L. masc. adj. equifaecalis, associated with the faeces of horses)	
A bacterial species identified by metagenomic analyses. This species includes all bacteria with genomes that show ≥95% average nucleotide identity (ANI) to the type genome for the species to which we have assigned the MAG ID E5_MB2_108 and which is available via NCBI BioSample SAMN18472541. The GC content of the type genome is 51.7% and the genome length is 1.71 Mbp.	
Description of Candidatus Cacconaster merdequi sp. nov.	
Candidatus Cacconaster merdequi (merd.e’qui. L. fem. n. merda, faeces; L. masc. n. equus, a horse; N.L. gen. n. merdequi, associated with the faeces of horses)	
A bacterial species identified by metagenomic analyses. This species includes all bacteria with genomes that show ≥95% average nucleotide identity (ANI) to the type genome for the species to which we have assigned the MAG ID E5_MB2_33 and which is available via NCBI BioSample SAMN18472547. The GC content of the type genome is 49% and the genome length is 1.90 Mbp.	
Description of Candidatus Cacconaster scatequi sp. nov.	
Candidatus Cacconaster scatequi (scat.e’qui. Gr. neut. n. skor, skatos, dung; L. masc. n. equus, a horse; N.L. gen. n. scatequi, associated with the faeces of horses)	
A bacterial species identified by metagenomic analyses. This species includes all bacteria with genomes that show ≥95% average nucleotide identity (ANI) to the type genome for the species to which we have assigned the MAG ID E3_MB2_97 and which is available via NCBI BioSample SAMN18472499. The GC content of the type genome is 50.6% and the genome length is 1.90 Mbp.	
Description of Candidatus Cacconaster stercorequi sp. nov.	
Candidatus Cacconaster stercorequi (ster.cor.e’qui. L. masc. n. stercus, stercoris, dung; L. masc. n. equus, a horse; N.L. gen. n. stercorequi, associated with the faeces of horses)	
A bacterial species identified by metagenomic analyses. This species includes all bacteria with genomes that show ≥95% average nucleotide identity (ANI) to the type genome for the species to which we have assigned the MAG ID E4_MB2_17 and which is available via NCBI BioSample SAMN18472518. The GC content of the type genome is 54.5% and the genome length is 1.83 Mbp.	
Description of Candidatus Chryseobacterium enterohippi sp. nov.	
Candidatus Chryseobacterium enterohippi (en.te.ro.hip’pi. Gr. neut. n. enteron, gut, bowel, intestine; Gr. masc./fem. n. hippos, a horse; N.L. gen. n. enterohippi, associated with the horse gut)	
A bacterial species identified by metagenomic analyses. This species includes all bacteria with genomes that show ≥95% average nucleotide identity (ANI) to the type genome for the species to which we have assigned the MAG ID E1_189 and which is available via NCBI BioSample SAMN18472455. The GC content of the type genome is 34.3% and the genome length is 2.05 Mbp.	
Description of Candidatus Colenecus gen. nov.	
Candidatus Colenecus (Col.en.e’cus. L. neut. n. colon, large intestine; N.L. masc. n. enecus, an inhabitant; N.L. masc. n. Colenecus, a microbe associated with the large intestine)	
A bacterial genus identified by metagenomic analyses. The genus includes all bacteria with genomes that show ≥60% average amino acid identity (AAI) to the genome of the type strain from the type species Candidatus Colenecus caballi. This is a new name for the GTDB alphanumeric genus UBA1179, which is found in diverse mammalian guts. This genus has been assigned by GTDB-Tk v1.3.0 working on GTDB Release 06-RS202 (Olm et al., 2017; Scales et al., 2018) to the order Bacteroidales and to the family Bacteroidaceae.	
Description of Candidatus Colenecus caballi sp. nov.	
Candidatus Colenecus caballi (ca.bal’li. L. gen. masc. n. caballi, of a horse)	
A bacterial species identified by metagenomic analyses. This species includes all bacteria with genomes that show ≥95% average nucleotide identity (ANI) to the type genome for the species to which we have assigned the MAG ID E3_160 and which is available via NCBI BioSample SAMN18472483. The GC content of the type genome is 49.7% and the genome length is 2.25 Mbp.	
Description of Candidatus Colicola gen. nov.	
Candidatus Colicola (Co.li.co’la. L. neut. n. colon, large intestine; N.L. masc./fem. suffix –cola, an inhabitant; N.L. fem. n. Colicola, a microbe associated with the large intestine)	
A bacterial genus identified by metagenomic analyses. The genus includes all bacteria with genomes that show ≥60% average amino acid identity (AAI) to the genome of the type strain from the type species Candidatus Colicola caballi. This is a new name for the GTDB alphanumeric genus RF16, which is found in diverse mammalian guts. This genus has been assigned by GTDB-Tk v1.3.0 working on GTDB Release 06-RS202 (Olm et al., 2017; Scales et al., 2018) to the order Bacteroidales and to the family Paludibacteraceae.	
Description of Candidatus Colicola caballi sp. nov.	
Candidatus Colicola caballi (ca.bal’li. L. gen. masc. n. caballi, of a horse)	
A bacterial species identified by metagenomic analyses. This species includes all bacteria with genomes that show ≥95% average nucleotide identity (ANI) to the type genome for the species to which we have assigned the MAG ID E1_MB2_58 and which is available via NCBI BioSample SAMN18472465. The GC content of the type genome is 46.6% and the genome length is 1.45 Mbp.	
Description of Candidatus Colicola caccequi sp. nov.	
Candidatus Colicola caccequi (cacc.e’qui. Gr. fem. n. kakke, faeces; L. masc. n. equus, a horse; N.L. gen. n. caccequi, associated with the faeces of horses)	
A bacterial species identified by metagenomic analyses. This species includes all bacteria with genomes that show ≥95% average nucleotide identity (ANI) to the type genome for the species to which we have assigned the MAG ID E4_134 and which is available via NCBI BioSample SAMN18472502. The GC content of the type genome is 44.2% and the genome length is 1.71 Mbp.	
Description of Candidatus Colicola coprequi sp. nov.	
Candidatus Colicola coprequi (copr.e’qui. Gr. fem. n. kopros, dung; L. masc. n. equus, a horse; N.L. gen. n. coprequi, associated with the faeces of horses)	
A bacterial species identified by metagenomic analyses. This species includes all bacteria with genomes that show ≥95% average nucleotide identity (ANI) to the type genome for the species to which we have assigned the MAG ID E2_MB2_30 and which is available via NCBI BioSample SAMN18472476. The GC content of the type genome is 46.1% and the genome length is 1.53 Mbp.	
Description of Candidatus Colicola equi sp. nov.	
Candidatus Colicola equi (e’qui. L. gen. masc. n. equi, of a horse)	
A bacterial species identified by metagenomic analyses. This species includes all bacteria with genomes that show ≥95% average nucleotide identity (ANI) to the type genome for the species to which we have assigned the MAG ID E1_186 and which is available via NCBI BioSample SAMN18472454. The GC content of the type genome is 44.4% and the genome length is 2.05 Mbp.	
Description of Candidatus Colicola faecequi sp. nov.	
Candidatus Colicola faecequi (faec.e’qui. L. fem. n. faex, faeces, dregs; L. masc. n. equus, a horse; N.L. gen. n. faecequi, associated with the faeces of horses)	
A bacterial species identified by metagenomic analyses. This species includes all bacteria with genomes that show ≥95% average nucleotide identity (ANI) to the type genome for the species to which we have assigned the MAG ID E4_MB2_124 and which is available via NCBI BioSample SAMN18472515. The GC content of the type genome is 52.3% and the genome length is 1.86 Mbp.	
Description of Candidatus Colimonas gen. nov.	
Candidatus Colimonas (Co.li.mo’nas. L. neut. n. colon, large intestine; L. fem. n. monas, a monad; N.L. fem. n. Colimonas, a microbe associated with the large intestine)	
A bacterial genus identified by metagenomic analyses. The genus includes all bacteria with genomes that show ≥60% average amino acid identity (AAI) to the genome of the type strain from the type species Candidatus Colimonas fimequi. This is a new name for the GTDB alphanumeric genus UBA1191, which is found in diverse mammalian guts. This genus has been assigned by GTDB-Tk v1.3.0 working on GTDB Release 06-RS202 (Olm et al., 2017; Scales et al., 2018) to the order Peptostreptococcales and to the family Anaerovoracaceae.	
Description of Candidatus Colimonas fimequi sp. nov.	
Candidatus Colimonas fimequi (fim.e’qui. L. masc. n. fimus, dung; L. masc. n. equus, a horse; N.L. gen. n. fimequi, associated with the faeces of horses)	
A bacterial species identified by metagenomic analyses. This species includes all bacteria with genomes that show ≥95% average nucleotide identity (ANI) to the type genome for the species to which we have assigned the MAG ID E4_13 and which is available via NCBI BioSample SAMN18472501. The GC content of the type genome is 44.3% and the genome length is 1.70 Mbp.	
Description of Candidatus Colimorpha gen. nov.	
Candidatus Colimorpha (Co.li.mor’pha. L. neut. n. colon, large intestine; Gr. fem. n. morphe, a form, shape; N.L. fem. n. Colimorpha, a microbe associated with the large intestine)	
A bacterial genus identified by metagenomic analyses. The genus includes all bacteria with genomes that show ≥60% average amino acid identity (AAI) to the genome of the type strain from the type species Candidatus Colimorpha merdihippi. This is a new name for the GTDB alphanumeric genus UBA1711, which is found in diverse mammalian guts. This genus has been assigned by GTDB-Tk v1.3.0 working on GTDB Release 06-RS202 (Olm et al., 2017; Scales et al., 2018) to the order Bacteroidales and to the family P3.	
Description of Candidatus Colimorpha enterica sp. nov.	
Candidatus Colimorpha enterica (en.te’ri.ca. Gr. neut. n. enteron, gut, bowel, intestine; L. fem. adj. suff. -ica, pertaining to; N.L. fem. adj. enterica, pertaining to intestine)	
A bacterial species identified by metagenomic analyses. This species includes all bacteria with genomes that show ≥95% average nucleotide identity (ANI) to the type genome for the species to which we have assigned the MAG ID E3_60 and which is available via NCBI BioSample SAMN18472488. This is a new name for the alphanumeric GTDB species sp000433515, which is found in diverse mammalian guts. The GC content of the type genome is 52.3% and the genome length is 1.43 Mbp.	
Description of Candidatus Colimorpha merdihippi sp. nov.	
Candidatus Colimorpha merdihippi (mer.di.hip’pi. L. fem. n. merda, faeces; Gr. masc./fem. n. hippos, a horse; N.L. gen. n. merdihippi, associated with the faeces of horses)	
A bacterial species identified by metagenomic analyses. This species includes all bacteria with genomes that show ≥95% average nucleotide identity (ANI) to the type genome for the species to which we have assigned the MAG ID E1_90 and which is available via NCBI BioSample SAMN18472457. The GC content of the type genome is 48.5% and the genome length is 3.11 Mbp.	
Description of Candidatus Colimorpha onthohippi sp. nov.	
Candidatus Colimorpha onthohippi (on.tho.hip’pi. Gr. masc. n. onthos, dung; Gr. masc./fem. n. hippos, a horse; N.L. gen. n. onthohippi, associated with the faeces of horses)	
A bacterial species identified by metagenomic analyses. This species includes all bacteria with genomes that show ≥95% average nucleotide identity (ANI) to the type genome for the species to which we have assigned the MAG ID E5_36 and which is available via NCBI BioSample SAMN18472537. The GC content of the type genome is 46.2% and the genome length is 2.04 Mbp.	
Description of Candidatus Colimorpha pelethequi sp. nov.	
Candidatus Colimorpha pelethequi (pe.leth.e’qui. Gr. masc. n. pelethos, dung; L. masc. n. equus, a horse; N.L. gen. n. pelethequi, associated with the faeces of horses)	
A bacterial species identified by metagenomic analyses. This species includes all bacteria with genomes that show ≥95% average nucleotide identity (ANI) to the type genome for the species to which we have assigned the MAG ID E5_MB2_81 and which is available via NCBI BioSample SAMN18472551. The GC content of the type genome is 46.7% and the genome length is 2.38 Mbp.	
Description of Candidatus Colinaster gen. nov.	
Candidatus Colinaster (Co.li.nas’ter. L. neut. n. colon, large intestine; Gr. masc. n. naster, an inhabitant; N.L. masc. n. Colinaster a microbe associated with the large intestine)	
A bacterial genus identified by metagenomic analyses. The genus includes all bacteria with genomes that show ≥60% average amino acid identity (AAI) to the genome of the type strain from the type species Candidatus Colinaster scatohippi. This is a new name for the GTDB alphanumeric genus UBA1712, which is found in diverse mammalian guts. This genus has been assigned by GTDB-Tk v1.3.0 working on GTDB Release 06-RS202 (Olm et al., 2017; Scales et al., 2018) to the order Lachnospirales and to the family Lachnospiraceae.	
Description of Candidatus Colinaster equi sp. nov.	
Candidatus Colinaster equi sp. nov.	
(e’qui. L. gen. masc. n. equi, of a horse)	
A bacterial species identified by metagenomic analyses. This species includes all bacteria with genomes that show ≥95% average nucleotide identity (ANI) to the type genome for the species to which we have assigned the MAG ID E5_MB2_109 and which is available via NCBI BioSample SAMN18472522. The GC content of the type genome is 39.2% and the genome length is 2.33 Mbp.	
Description of Candidatus Colinaster scatohippi sp. nov.	
Candidatus Colinaster scatohippi (sca.to.hip’pi. Gr. neut. n. skor, skatos, dung; Gr. masc./fem. n. hipposa horse; N.L. gen. n. scatohippi, associated with the faeces of horses)	
A bacterial species identified by metagenomic analyses. This species includes all bacteria with genomes that show ≥95% average nucleotide identity (ANI) to the type genome for the species to which we have assigned the MAG ID E4_MB2_45 and which is available via NCBI BioSample SAMN18472524. The GC content of the type genome is 38.7% and the genome length is 2.18 Mbp.	
Description of Candidatus Coliplasma gen. nov.	
Candidatus Coliplasma (Co.li.plas’ma. L. neut. n. colon, large intestine; Gr. neut. n. plasma, a form; N.L. neut. n. Coliplasma, a microbe associated with the large intestine)	
A bacterial genus identified by metagenomic analyses. The genus includes all bacteria with genomes that show ≥60% average amino acid identity (AAI) to the genome of the type strain from the type species Candidatus Coliplasma caballi. This is a new name for the GTDB alphanumeric genus UBA1752, which is found in diverse mammalian guts. This genus has been assigned by GTDB-Tk v1.3.0 working on GTDB Release 06-RS202 (Olm et al., 2017; Scales et al., 2018) to the order Oscillospirales and to the family CAG-382.	
Description of Candidatus Coliplasma caballi sp. nov.	
Candidatus Coliplasma caballi (ca.bal’li. L. gen. masc. n. caballi, of a horse)	
A bacterial species identified by metagenomic analyses. This species includes all bacteria with genomes that show ≥95% average nucleotide identity (ANI) to the type genome for the species to which we have assigned the MAG ID E3_MB2_28 and which is available via NCBI BioSample SAMN18472492. The GC content of the type genome is 54.8% and the genome length is 1.41 Mbp.	
Description of Candidatus Coliplasma equi sp. nov.	
Candidatus Coliplasma equi (e’qui. L. gen. masc. n. equi, of a horse)	
A bacterial species identified by metagenomic analyses. This species includes all bacteria with genomes that show ≥95% average nucleotide identity (ANI) to the type genome for the species to which we have assigned the MAG ID E3_142 and which is available via NCBI BioSample SAMN18472481. The GC content of the type genome is 49.7% and the genome length is 1.52 Mbp.	
Description of Candidatus Colisoma gen. nov.	
Candidatus Colisoma (Co.li.so’ma. L. neut. n. colon, large intestine; Gr. neut. n. soma, a body; N.L. neut. n. Colisoma, a microbe associated with the large intestine)	
A bacterial genus identified by metagenomic analyses. The genus includes all bacteria with genomes that show ≥60% average amino acid identity (AAI) to the genome of the type strain from the type species Candidatus Colisoma equi. This is a new name for the GTDB alphanumeric genus UBA1067, which is found in diverse mammalian guts. This genus has been assigned by GTDB-Tk v1.3.0 working on GTDB Release 06-RS202 (Olm et al., 2017; Scales et al., 2018) to the order RFP12 and to the family UBA1067.	
Description of Candidatus Colisoma equi sp. nov.	
Candidatus Colisoma equi (e’qui. L. gen. masc. n. equi, of a horse)	
A bacterial species identified by metagenomic analyses. This species includes all bacteria with genomes that show ≥95% average nucleotide identity (ANI) to the type genome for the species to which we have assigned the MAG ID E4_MB2_14 and which is available via NCBI BioSample SAMN18472517. The GC content of the type genome is 60% and the genome length is 2.52 Mbp.	
Description of Candidatus Colivicinus gen. nov.	
Candidatus Colivicinus (Co.li.vi’ci.nus. L. neut. n. colon, large intestine; N.L. masc. n. vicinus, a neighbour; N.L. masc. n. Colivicinus, a microbe associated with the large intestine)	
A bacterial genus identified by metagenomic analyses. The genus includes all bacteria with genomes that show ≥60% average amino acid identity (AAI) to the genome of the type strain from the type species Candidatus Colivicinus equi. This is a new name for the GTDB alphanumeric genus UBA636, which is found in diverse mammalian guts. This genus has been assigned by GTDB-Tk v1.3.0 working on GTDB Release 06-RS202 (Olm et al., 2017; Scales et al., 2018) to the order Erysipelotrichales and to the family Erysipelotrichaceae.	
Description of Candidatus Colivicinus equi sp. nov.	
Candidatus Colivicinus equi (e’qui. L. gen. masc. n. equi, of a horse)	
A bacterial species identified by metagenomic analyses. This species includes all bacteria with genomes that show ≥95% average nucleotide identity (ANI) to the type genome for the species to which we have assigned the MAG ID E4_MB2_36 and which is available via NCBI BioSample SAMN18472522. The GC content of the type genome is 31.9% and the genome length is 1.69 Mbp.	
Description of Candidatus Colivivens gen. nov.	
Candidatus Colivivens (Co.li.vi’vens. L. neut. n. colon, large intestine; N.L. masc./fem. pres. part. vivens, living; N.L. fem. n. Colivivens, a microbe associated with the large intestine)	
A bacterial genus identified by metagenomic analyses. The genus includes all bacteria with genomes that show ≥60% average amino acid identity (AAI) to the genome of the type strain from the type species Candidatus Colivivens caballi. This is a new name for the GTDB alphanumeric genus UBA1786, which is found in diverse mammalian guts. This genus has been assigned by GTDB-Tk v1.3.0 working on GTDB Release 06-RS202 (Olm et al., 2017; Scales et al., 2018) to the order Bacteroidales and to the family Bacteroidaceae.	
Description of Candidatus Colivivens caballi sp. nov.	
Candidatus Colivivens caballi (ca.bal’li. L. gen. masc. n. caballi, of a horse)	
A bacterial species identified by metagenomic analyses. This species includes all bacteria with genomes that show ≥95% average nucleotide identity (ANI) to the type genome for the species to which we have assigned the MAG ID E3_198 and which is available via NCBI BioSample SAMN18472484. The GC content of the type genome is 47.7% and the genome length is 2.55 Mbp.	
Description of Candidatus Colivivens equi sp. nov.	
Candidatus Colivivens equi (e’qui. L. gen. masc. n. equi, of a horse)	
A bacterial species identified by metagenomic analyses. This species includes all bacteria with genomes that show ≥95% average nucleotide identity (ANI) to the type genome for the species to which we have assigned the MAG ID E1_MB2_52 and which is available via NCBI BioSample SAMN18472463. The GC content of the type genome is 38.2% and the genome length is 2.64 Mbp.	
Description of Candidatus Colousia gen. nov.	
Candidatus Colousia (Col.ou’s.ia. L. neut. n. colon, large intestine; Gr. fem. n. ousia, an essence; N.L. fem. n. Colousia, a microbe associated with the large intestine)	
A bacterial genus identified by metagenomic analyses. The genus includes all bacteria with genomes that show ≥60% average amino acid identity (AAI) to the genome of the type strain from the type species Candidatus Colousia faecequi. This is a new name for the GTDB alphanumeric genus SFVR01, which is found in diverse mammalian guts. This genus has been assigned by GTDB-Tk v1.3.0 working on GTDB Release 06-RS202 (Olm et al., 2017; Scales et al., 2018) to the order Bacteroidales and to the family Paludibacteraceae.	
Description of Candidatus Colousia faecequi sp. nov.	
Candidatus Colousia faecequi (faec.e’qui. L. fem. n. faex, faeces, dregs; L. masc. n. equus, a horse; N.L. gen. n. faecequi, associated with the faeces of horses)	
A bacterial species identified by metagenomic analyses. This species includes all bacteria with genomes that show ≥95% average nucleotide identity (ANI) to the type genome for the species to which we have assigned the MAG ID E3_MB2_91 and which is available via NCBI BioSample SAMN18472498. The GC content of the type genome is 47.1% and the genome length is 1.67 Mbp.	
Description of Candidatus Comamonas equi sp. nov.	
Candidatus Comamonas equi (e’qui. L. gen. masc. n. equi, of a horse)	
A bacterial species identified by metagenomic analyses. This species includes all bacteria with genomes that show ≥95% average nucleotide identity (ANI) to the type genome for the species to which we have assigned the MAG ID E2_118 and which is available via NCBI BioSample SAMN18472472. The GC content of the type genome is 59.2% and the genome length is 2.60 Mbp.	
Description of Candidatus Copronaster gen. nov.	
Candidatus Copronaster (Co.pro.nas’ter. Gr. fem. n. kopros, dung; Gr. masc. n. naster, an inhabitant; N.L. masc. n. Copronaster, a microbe associated with faeces)	
A bacterial genus identified by metagenomic analyses. The genus includes all bacteria with genomes that show ≥60% average amino acid identity (AAI) to the genome of the type strain from the type species Candidatus Copronaster equi. This is a new name for the GTDB alphanumeric genus CAG-488, which is found in diverse mammalian guts. This genus has been assigned by GTDB-Tk v1.3.0 working on GTDB Release 06-RS202 (Olm et al., 2017; Scales et al., 2018) to the order Oscillospirales and to the family Acutalibacteraceae.	
Description of Candidatus Copronaster equi sp. nov.	
Candidatus Copronaster equi (e’qui. L. gen. masc. n. equi, of a horse)	
A bacterial species identified by metagenomic analyses. This species includes all bacteria with genomes that show ≥95% average nucleotide identity (ANI) to the type genome for the species to which we have assigned the MAG ID E5_MB2_59 and which is available via NCBI BioSample SAMN18472549. The GC content of the type genome is 39.1% and the genome length is 1.98 Mbp.	
Description of Candidatus Crickella gen. nov.	
Candidatus Crickella gen. nov. (Cric’kel.la N.L. fem. dim. n. Crickella, named in honour of Francis Crick, the British molecular biologist who played a crucial roles in deciphering the helical structure of the DNA molecule). A bacterial genus identified by metagenomic analyses. The genus includes all bacteria with genomes that show ≥60% average amino acid identity (AAI) to the genome of the type strain from the type species Candidatus Crickella caballi. This is a new name for the GTDB alphanumeric genus RUG099, which is found in diverse mammalian guts. This genus has been assigned by GTDB-Tk v1.3.0 working on GTDB Release 06-RS202 (Olm et al., 2017; Scales et al., 2018) to the order Peptostreptococcales and to the family Anaerovoracaceae.	
Description of Candidatus Crickella caballi sp. nov.	
Candidatus Crickella caballi sp. nov. (ca.bal’li. L. gen. masc. n. caballi, of a horse)	
A bacterial species identified by metagenomic analyses. This species includes all bacteria with genomes that show ≥95% average nucleotide identity (ANI) to the type genome for the species to which we have assigned the MAG ID E4_MB2_90 and which is available via NCBI BioSample SAMN18472532. The GC content of the type genome is 45.2% and the genome length is 1.45 Mbp.	
Description of Candidatus Crickella equi sp. nov.	
Candidatus Crickella equi sp. nov. (e’qui. L. gen. masc. n. equi, of a horse)	
A bacterial species identified by metagenomic analyses. This species includes all bacteria with genomes that show ≥95% average nucleotide identity (ANI) to the type genome for the species to which we have assigned the MAG ID E4_MB2_51 and which is available via NCBI BioSample SAMN18472526. The GC content of the type genome is 43.4% and the genome length is 1.39 Mbp.	
Description of Candidatus Crickella merdequi sp. nov.	
Candidatus Crickella merdequi (merd.e’qui. L. fem. n. merda, faeces; L. masc. n. equus, a horse; N.L. gen. n. merdequi, associated with the faeces of horses)	
A bacterial species identified by metagenomic analyses. This species includes all bacteria with genomes that show ≥95% average nucleotide identity (ANI) to the type genome for the species to which we have assigned the MAG ID E4_MB2_84 and which is available via NCBI BioSample SAMN18472530. The GC content of the type genome is 43.8% and the genome length is 1.61 Mbp.	
Description of Candidatus Cryptobacteroides aphodequi sp. nov.	
Candidatus Cryptobacteroides aphodequi (aph.od.e’qui. Gr. fem. n. aphodos, dung; L. masc. n. equus, a horse; N.L. gen. n. aphodequi, associated with the faeces of horses)	
A bacterial species identified by metagenomic analyses. This species includes all bacteria with genomes that show ≥95% average nucleotide identity (ANI) to the type genome for the species to which we have assigned the MAG ID E3_MB2_98 and which is available via NCBI BioSample SAMN18472500. The GC content of the type genome is 54.9% and the genome length is 1.48 Mbp.	
Description of Candidatus Cryptobacteroides caccocaballi sp. nov.	
Candidatus Cryptobacteroides caccocaballi (cac.co.ca.bal’li. Gr. fem. n. kakke, faeces; L. masc. n. caballus, a horse; N.L. gen. n. caccocaballi, associated with the faeces of horses)	
A bacterial species identified by metagenomic analyses. This species includes all bacteria with genomes that show ≥95% average nucleotide identity (ANI) to the type genome for the species to which we have assigned the MAG ID E4_MB2_58 and which is available via NCBI BioSample SAMN18472527. The GC content of the type genome is 51.8% and the genome length is 2.22 Mbp.	
Description of Candidatus Cryptobacteroides choladohippi sp. nov.	
Candidatus Cryptobacteroides choladohippi (cho.la.do.hip’pi. Gr. fem. n. kholas, kholados, guts; Gr. masc./fem. n. hipposa horse; N.L. gen. n. choladohippi, associated with the horse gut)	
A bacterial species identified by metagenomic analyses. This species includes all bacteria with genomes that show ≥95% average nucleotide identity (ANI) to the type genome for the species to which we have assigned the MAG ID E1_MB2_55 and which is available via NCBI BioSample SAMN18472464. The GC content of the type genome is 54.2% and the genome length is 2.24 Mbp.	
Description of Candidatus Cryptobacteroides equifaecalis sp. nov.	
Candidatus Cryptobacteroides equifaecalis (e.qui.fae.ca’lis. L. masc. n. equus, a horse; N.L. masc. adj. faecalis, faecal; N.L. masc. adj. equifaecalis, associated with the faeces of horses)	
A bacterial species identified by metagenomic analyses. This species includes all bacteria with genomes that show ≥95% average nucleotide identity (ANI) to the type genome for the species to which we have assigned the MAG ID E4_MB2_98 and which is available via NCBI BioSample SAMN18472533. The GC content of the type genome is 52.5% and the genome length is 1.61 Mbp.	
Description of Candidatus Cryptobacteroides faecihippi sp. nov.	
Candidatus Cryptobacteroides faecihippi (fae.ci.hip’pi. L. fem. n. faex, faeces, dregs; Gr. masc./fem. n. hipposa horse; N.L. gen. n. faecihippi, associated with the faeces of horses)	
A bacterial species identified by metagenomic analyses. This species includes all bacteria with genomes that show ≥95% average nucleotide identity (ANI) to the type genome for the species to which we have assigned the MAG ID E1_MB2_112 and which is available via NCBI BioSample SAMN18472461. The GC content of the type genome is 54.8% and the genome length is 2.25 Mbp.	
Description of Candidatus Cryptobacteroides fimicaballi sp. nov.	
Candidatus Cryptobacteroides fimicaballi (fi.mi.ca.bal’li. L. masc. n. fimus, dung; L. masc. n. caballus, a horse; N.L. gen. n. fimicaballi, associated with the faeces of horses)	
A bacterial species identified by metagenomic analyses. This species includes all bacteria with genomes that show ≥95% average nucleotide identity (ANI) to the type genome for the species to which we have assigned the MAG ID E3_MB2_135 and which is available via NCBI BioSample SAMN18472490. The GC content of the type genome is 51% and the genome length is 1.33 Mbp.	
Description of Candidatus Cryptobacteroides onthequi sp. nov.	
Candidatus Cryptobacteroides onthequi (onth.e’qui. Gr. masc. n. onthos, dung; L. masc. n. equus, a horse; N.L. gen. n. onthequi, associated with the faeces of horses)	
A bacterial species identified by metagenomic analyses. This species includes all bacteria with genomes that show ≥95% average nucleotide identity (ANI) to the type genome for the species to which we have assigned the MAG ID E1_MB2_10 and which is available via NCBI BioSample SAMN18472459. The GC content of the type genome is 53.4% and the genome length is 2.96 Mbp.	
Description of Candidatus Darwinibacterium gen. nov.	
Candidatus Darwinibacterium gen. nov.	
(Dar.win.i.bac.te’ri.um N.L. masc. n. darwinii derived from the Latinised family name of Charles Darwin; N.L. neut. n. bacterium, a small rod or staff; N.L. neut. n. Darwinibacterium, a microbe named in honour of Charles Darwin, British scientist who proposed the theory of evolution by natural selection). A bacterial genus identified by metagenomic analyses. The genus includes all bacteria with genomes that show ≥60% average amino acid identity (AAI) to the genome of the type strain from the type species Candidatus Darwinibacterium equi. This genus has been assigned by GTDB-Tk v1.3.0 working on GTDB Release 06-RS202 (Olm et al., 2017; Scales et al., 2018) to the order Oscillospirales and to the family CAG-272.	
Description of Candidatus Darwinibacterium equi sp. nov.	
Candidatus Darwinibacterium equi sp. nov.	
(e’qui. L. gen. masc. n. equi, of a horse)	
A bacterial species identified by metagenomic analyses. This species includes all bacteria with genomes that show ≥95% average nucleotide identity (ANI) to the type genome for the species to which we have assigned the MAG ID E2_100 and which is available via NCBI BioSample SAMN18472522. The GC content of the type genome is 49.5% and the genome length is 1.54 Mbp.	
Description of Candidatus Darwinimomas gen. nov.	
Candidatus Darwinimomas gen. nov.	
(Dar.win.i.mo.nas.N.L. masc. n. darwini derived from the Latinised family name of Charles Darwin; L. fem. n. monas, unit, monad; N.L. fem n. Darwinimomas, a microbe named in honour of Charles Darwin, British scientist who proposed the theory of evolution by natural selection). A bacterial genus identified by metagenomic analyses. The genus includes all bacteria with genomes that show ≥60% average amino acid identity (AAI) to the genome of the type strain from the type species Candidatus Darwinimomas equi. This is a new name for the GTDB alphanumeric genus UBA1755, which is found in diverse mammalian guts. This genus has been assigned by GTDB-Tk v1.3.0 working on GTDB Release 06-RS202 (Olm et al., 2017; Scales et al., 2018) to the order Lachnospirales and to the family Lachnospiraceae.	
Description of Candidatus Darwinimomas equi sp. nov.	
Candidatus Darwinimomas equi sp. nov. (e’qui. L. gen. masc. n. equi, of a horse)	
A bacterial species identified by metagenomic analyses. This species includes all bacteria with genomes that show ≥95% average nucleotide identity (ANI) to the type genome for the species to which we have assigned the MAG ID E1_MB2_36 and which is available via NCBI BioSample SAMN18472462. GTDB has assigned this species to a genus marked with an alphabetical suffix. However, as this genus designation cannot be incorporated into a well-formed binomial, in naming. this species, we have used the current validly published name for the genus. The GC content of the type genome is 44.2% and the genome length is 1.44 Mbp.	
Description of Candidatus Egerieousia equi sp. nov.	
Candidatus Egerieousia equi (e’qui. L. gen. masc. n. equi, of a horse)	
A bacterial species identified by metagenomic analyses. This species includes all bacteria with genomes that show ≥95% average nucleotide identity (ANI) to the type genome for the species to which we have assigned the MAG ID E4_MB2_106 and which is available via NCBI BioSample SAMN18472513. The GC content of the type genome is 46.4% and the genome length is 1.92 Mbp.	
Description of Candidatus Enterousia merdequi sp. nov.	
Candidatus Enterousia merdequi (merd.e’qui. L. fem. n. merda, faeces; L. masc. n. equus, a horse; N.L. gen. n. merdequi, associated with the faeces of horses)	
A bacterial species identified by metagenomic analyses. This species includes all bacteria with genomes that show ≥95% average nucleotide identity (ANI) to the type genome for the species to which we have assigned the MAG ID E3_MB2_90 and which is available via NCBI BioSample SAMN18472497. The GC content of the type genome is 33.9% and the genome length is 0.74 Mbp.	
Description of Candidatus Enterousia onthequi sp. nov.	
Candidatus Enterousia onthequi (onth.e’qui. Gr. masc. n. onthos, dung; L. masc. n. equus, a horse; N.L. gen. n. onthequi, associated with the faeces of horses)	
A bacterial species identified by metagenomic analyses. This species includes all bacteria with genomes that show ≥95% average nucleotide identity (ANI) to the type genome for the species to which we have assigned the MAG ID E5_MB2_19 and which is available via NCBI BioSample SAMN18472546. The GC content of the type genome is 38.8% and the genome length is 0.88 Mbp.	
Description of Candidatus Enterousia scatequi sp. nov.	
Candidatus Enterousia scatequi (scat.e’qui. Gr. neut. n. skor, skatos, dung; L. masc. n. equus, a horse; N.L. gen. n. scatequi, associated with the faeces of horses)	
A bacterial species identified by metagenomic analyses. This species includes all bacteria with genomes that show ≥95% average nucleotide identity (ANI) to the type genome for the species to which we have assigned the MAG ID E5_MB2_120 and which is available via NCBI BioSample SAMN18472543. The GC content of the type genome is 39.9% and the genome length is 0.76 Mbp.	
Description of Candidatus Equadaptatus gen. nov.	
Candidatus Equadaptaus (Equ.a.dap.ta’tus. L. masc. n. equus, a horse; L. masc. perf. part. adaptatus, adapted to; N.L. masc. n. Equiadaptatus, a microbe associated with horses)	
A bacterial genus identified by metagenomic analyses. The genus includes all bacteria with genomes that show ≥60% average amino acid identity (AAI) to the genome of the type strain from the type species Candidatus Equadaptatus faecalis. This genus has been assigned by GTDB-Tk v1.3.0 working on GTDB Release 06-RS202 (Olm et al., 2017; Scales et al., 2018) to the order Synergistales and to the family Synergistaceae.	
Description of Candidatus Equadaptatus faecalis sp. nov.	
Candidatus Equadaptatus faecalis (fae.ca’lis. N.L. masc. adj. faecalis, faecal)	
A bacterial species identified by metagenomic analyses. This species includes all bacteria with genomes that show ≥95% average nucleotide identity (ANI) to the type genome for the species to which we have assigned the MAG ID E4_60 and which is available via NCBI BioSample SAMN18472510. The GC content of the type genome is 48.4% and the genome length is 1.60 Mbp.	
Description of Candidatus Equibacterium gen. nov.	
Candidatus Equibacterium (E.qui.bac.te’ri.um. L. masc. n. equus, a horse; L. neut. n. bacterium, a bacterium; N.L. neut. n. Equibacterium a microbe associated with horses)	
A bacterial genus identified by metagenomic analyses. The genus includes all bacteria with genomes that show ≥60% average amino acid identity (AAI) to the genome of the type strain from the type species Candidatus Equibacterium intestinale. This genus has been assigned by GTDB-Tk v1.3.0 working on GTDB Release 06-RS202 (Olm et al., 2017; Scales et al., 2018) to the order Bacteroidales and to the family UBA932.	
Description of Candidatus Equibacterium intestinale sp. nov.	
Candidatus Equibacterium intestinale (in.tes.ti.na’le.N.L. neut. adj. intestinale, pertaining to the intestines)	
A bacterial species identified by metagenomic analyses. This species includes all bacteria with genomes that show ≥95% average nucleotide identity (ANI) to the type genome for the species to which we have assigned the MAG ID E5_MB2_82 and which is available via NCBI BioSample SAMN18472552. The GC content of the type genome is 52.3% and the genome length is 1.76 Mbp.	
Description of Candidatus Equicaccousia gen. nov.	
Candidatus Equicaccousia (E.qui.cacc.ou’s.ia. L. masc. n. equus, a horse; Gr. fem. n. kakke, faeces; Gr. fem. n. ousia, an essence; N.L. fem. n. Equicaccousia, a microbe associated with horse faeces)	
A bacterial genus identified by metagenomic analyses. The genus includes all bacteria with genomes that show ≥60% average amino acid identity (AAI) to the genome of the type strain from the type species Candidatus Equicaccousia limihippi. This is a new name for the GTDB alphanumeric genus UMGS1279, which is found in diverse mammalian guts. This genus has been assigned by GTDB-Tk v1.3.0 working on GTDB Release 06-RS202 (Olm et al., 2017; Scales et al., 2018) to the order Oscillospirales and to the family Acutalibacteraceae.	
Description of Candidatus Equicaccousia limihippi sp. nov.	
Candidatus Equicaccousia limihippi (li.mi.hip’pi. L. masc. n. limus, dung; Gr. masc./fem. n. hipposa horse; N.L. gen. n. limihippi, of horse dung)	
A bacterial species identified by metagenomic analyses. This species includes all bacteria with genomes that show ≥95% average nucleotide identity (ANI) to the type genome for the species to which we have assigned the MAG ID E2_98 and which is available via NCBI BioSample SAMN18472475. The GC content of the type genome is 44.9% and the genome length is 1.15 Mbp.	
Description of Candidatus Equicola gen. nov.	
Candidatus Equicola (E.qui’co.la. L. masc. n. equus, a horse; N.L. masc./fem. suffix –cola, an inhabitant; N.L. fem. n. Equicola, a microbe associated with horses)	
A bacterial genus identified by metagenomic analyses. The genus includes all bacteria with genomes that show ≥60% average amino acid identity (AAI) to the genome of the type strain from the type species Candidatus Equicola stercoris. This genus has been assigned by GTDB-Tk v1.3.0 working on GTDB Release 06-RS202 (Olm et al., 2017; Scales et al., 2018) to the order Bacteroidales and to the family Bacteroidaceae.	
Description of Candidatus Equicola faecalis sp. nov.	
Candidatus Equicola faecalis (fae.ca’lis. N.L. fem. adj. faecalis, faecal)	
A bacterial species identified by metagenomic analyses. This species includes all bacteria with genomes that show ≥95% average nucleotide identity (ANI) to the type genome for the species to which we have assigned the MAG ID E4_176 and which is available via NCBI BioSample SAMN18472505. The GC content of the type genome is 44.8% and the genome length is 2.09 Mbp.	
Description of Candidatus Equicola stercoris sp. nov.	
Candidatus Equicola stercoris (ster’co.ris. L. gen. masc. n. stercoris, of dung)	
A bacterial species identified by metagenomic analyses. This species includes all bacteria with genomes that show ≥95% average nucleotide identity (ANI) to the type genome for the species to which we have assigned the MAG ID E3_MB2_38 and which is available via NCBI BioSample SAMN18472493. The GC content of the type genome is 42% and the genome length is 1.75 Mbp.	
Description of Candidatus Equihabitans gen. nov.	
Candidatus Equihabitans (E.qui.ha’bi.tans. L. masc. n. equus, a horse; L. masc./fem. pres. part. habitans, an inhabitant; N.L. fem. n. Equihabitans, a microbe associated with horses)	
A bacterial genus identified by metagenomic analyses. The genus includes all bacteria with genomes that show ≥60% average amino acid identity (AAI) to the genome of the type strain from the type species Candidatus Equihabitans merdae. This genus has been assigned by GTDB-Tk v1.3.0 working on GTDB Release 06-RS202 (Olm et al., 2017; Scales et al., 2018) to the order Lachnospirales and to the family Lachnospiraceae.	
Description of Candidatus Equihabitans merdae sp. nov.	
Candidatus Equihabitans merdae (mer’dae. L. gen. fem. n. merdae, of faeces)	
A bacterial species identified by metagenomic analyses. This species includes all bacteria with genomes that show ≥95% average nucleotide identity (ANI) to the type genome for the species to which we have assigned the MAG ID E4_98 and which is available via NCBI BioSample SAMN18472512. The GC content of the type genome is 47% and the genome length is 1.86 Mbp.	
Description of Candidatus Equimonas gen. nov.	
Candidatus Equimonas (E.qui.mo’nas. L. masc. n. equus, a horse; L. fem. n. monas, a monad; N.L. fem. n. Equimonas, a microbe associated with horses)	
A bacterial genus identified by metagenomic analyses. The genus includes all bacteria with genomes that show ≥60% average amino acid identity (AAI) to the genome of the type strain from the type species Candidatus Equimonas enterica. This genus has been assigned by GTDB-Tk v1.3.0 working on GTDB Release 06-RS202 (Olm et al., 2017; Scales et al., 2018) to the order Bacteroidales and to the family Bacteroidaceae.	
Description of Candidatus Equimonas enterica sp. nov.	
Candidatus Equimonas enterica (en.te’ri.ca. Gr. neut. n. enteron, gut, bowel, intestine; L.. fem. adj. suff. -ica, pertaining to; N.L. fem. adj. enterica, pertaining to intestine)	
A bacterial species identified by metagenomic analyses. This species includes all bacteria with genomes that show ≥95% average nucleotide identity (ANI) to the type genome for the species to which we have assigned the MAG ID E1_145 and which is available via NCBI BioSample SAMN18472453. The GC content of the type genome is 55.9% and the genome length is 1.85 Mbp.	
Description of Candidatus Equimonas faecalis sp. nov.	
Candidatus Equimonas faecalis (fae.ca’lis. N.L. fem. adj. faecalis, faecal)	
A bacterial species identified by metagenomic analyses. This species includes all bacteria with genomes that show ≥95% average nucleotide identity (ANI) to the type genome for the species to which we have assigned the MAG ID E1_115 and which is available via NCBI BioSample SAMN18472452. The GC content of the type genome is 55.6% and the genome length is 2.59 Mbp.	
Description of Candidatus Equinaster gen. nov.	
Candidatus Equinaster (E.qui.nas’ter. L. masc. n. equus, a horse; Gr. masc. n. naster, an inhabitant; N.L. masc. n. Equinaster, a microbe associated with horses)	
A bacterial genus identified by metagenomic analyses. The genus includes all bacteria with genomes that show ≥60% average amino acid identity (AAI) to the genome of the type strain from the type species Candidatus Equinaster intestinalis. This genus has been assigned by GTDB-Tk v1.3.0 working on GTDB Release 06-RS202 (Olm et al., 2017; Scales et al., 2018) to the order Oscillospirales and to the family Acutalibacteraceae.	
Description of Candidatus Equinaster intestinalis sp. nov.	
Candidatus Equinaster intestinalis (in.tes.ti.na’lis. N.L. masc. adj. intestinalis, pertaining to the intestines)	
A bacterial species identified by metagenomic analyses. This species includes all bacteria with genomes that show ≥95% average nucleotide identity (ANI) to the type genome for the species to which we have assigned the MAG ID E3_MB2_43 and which is available via NCBI BioSample SAMN18472494. The GC content of the type genome is 43.4% and the genome length is 1.50 Mbp.	
Description of Candidatus Faecinaster gen. nov.	
Candidatus Faecinaster (Fae.ci.nas’ter. L. fem. n. faex, faecis, dregs; Gr. masc. n. naster, an inhabitant; N.L. masc. n. Faecinaster, a microbe associated with faeces)	
A bacterial genus identified by metagenomic analyses. The genus includes all bacteria with genomes that show ≥60% average amino acid identity (AAI) to the genome of the type strain from the type species Candidatus Faecinaster equi. This is a new name for the GTDB alphanumeric genus UBA6382, which is found in diverse mammalian guts. This genus has been assigned by GTDB-Tk v1.3.0 working on GTDB Release 06-RS202 (Olm et al., 2017; Scales et al., 2018) to the order Bacteroidales and to the family Bacteroidaceae.	
Description of Candidatus Faecinaster equi sp. nov.	
Candidatus Faecinaster equi (e’qui. L. gen. masc. n. equi, of a horse)	
A bacterial species identified by metagenomic analyses. This species includes all bacteria with genomes that show ≥95% average nucleotide identity (ANI) to the type genome for the species to which we have assigned the MAG ID E3_MB2_9 and which is available via NCBI BioSample SAMN18472496. The GC content of the type genome is 37.2% and the genome length is 3.36 Mbp.	
Description of Candidatus Fiminaster gen. nov.	
Candidatus Fiminaster (Fi.mi.nas’ter. L. neut. n. fimum, dung; Gr. masc. n. naster, an inhabitant; N.L. masc. n. Fiminaster, a microbe associated with faeces)	
A bacterial genus identified by metagenomic analyses. The genus includes all bacteria with genomes that show ≥60% average amino acid identity (AAI) to the genome of the type strain from the type species Candidatus Fiminaster equi. This is a new name for the GTDB alphanumeric genus UBA3207, which is found in diverse mammalian guts. This genus has been assigned by GTDB-Tk v1.3.0 working on GTDB Release 06-RS202 (Olm et al., 2017; Scales et al., 2018) to the order RFN20 and to the family CAG-826.	
Description of Candidatus Fiminaster equi sp. nov.	
Candidatus Fiminaster equi (e’qui. L. gen. masc. n. equi, of a horse)	
A bacterial species identified by metagenomic analyses. This species includes all bacteria with genomes that show ≥95% average nucleotide identity (ANI) to the type genome for the species to which we have assigned the MAG ID E4_MB2_69 and which is available via NCBI BioSample SAMN18472528. The GC content of the type genome is 34.5% and the genome length is 0.89 Mbp.	
Description of Candidatus Hennigella gen. nov.	
Candidatus Hennigella gen. nov.	
(N.L. fem. dim. n.Hennigella, named in honour of Willi Hennig, founder of phylogenetic systematics)	
A bacterial genus identified by metagenomic analyses. The genus includes all bacteria with genomes that show ≥60% average amino acid identity (AAI) to the genome of the type strain from the type species Candidatus Hennigella equi. This is a new name for the GTDB alphanumeric genus RUG11194 which is found in diverse mammalian guts. This genus has been assigned by GTDB-Tk v1.3.0 working on GTDB Release 06-RS202 (Olm et al., 2017; Scales et al., 2018) to the order Mycoplasmatales and to the family Mycoplasmoidaceae	
Description of Candidatus Hennigella equi sp. nov.	
Candidatus Hennigella equi sp. nov. (e’qui. L. gen. masc. n. equi, of a horse)	
A bacterial species identified by metagenomic analyses. This species includes all bacteria with genomes that show ≥95% average nucleotide identity (ANI) to the type genome for the species to which we have assigned the MAG ID E4_MB2_29 and which is available via NCBI BioSample SAMN18472521. The GC content of the type genome is 31.4% and the genome length is 0.64 Mbp.	
Description of Candidatus Hennigimonas gen. nov.	
Candidatus Hennigimonas gen. nov. (N.L. masc. n. hennigi derived from the Latinised family name of Willi Hennig; L. fem. n. monas, unit, monad; a microbe named in honour of Willi Hennig, founder of phylogenetic systematics)	
A bacterial genus identified by metagenomic analyses. The genus includes all bacteria with genomes that show ≥60% average amino acid identity (AAI) to the genome of the type strain from the type species Candidatus Hennigimonas equi. This genus has been assigned by GTDB-Tk v1.3.0 working on GTDB Release 06-RS202 (Olm et al., 2017; Scales et al., 2018) to the order Bacteroidales and to the family UBA932	
Description of Candidatus Hennigimonas equi sp. nov.	
Candidatus Hennigimonas equi sp. nov. (e’qui. L. gen. masc. n. equi, of a horse)	
A bacterial species identified by metagenomic analyses. This species includes all bacteria with genomes that show ≥95% average nucleotide identity (ANI) to the type genome for the species to which we have assigned the MAG ID E3_MB2_147 and which is available via NCBI BioSample SAMN18472491. The GC content of the type genome is 52% and the genome length is 1.47 Mbp.	
Description of Candidatus Hippenecus gen. nov.	
Candidatus Hippenecus (Hipp.en.e’cus. Gr. masc./fem. n. hippos, a horse; N.L. masc. n. enecus, an inhabitant; N.L. masc. n. Hippenecus a microbe associated with horses)	
A bacterial genus identified by metagenomic analyses. The genus includes all bacteria with genomes that show ≥60% average amino acid identity (AAI) to the genome of the type strain from the type species Candidatus Hippenecus merdae. This genus has been assigned by GTDB-Tk v1.3.0 working on GTDB Release 06-RS202 (Olm et al., 2017; Scales et al., 2018) to the order Lachnospirales and to the family Lachnospiraceae.	
Description of Candidatus Hippenecus merdae sp. nov.	
Candidatus Hippenecus merdae (mer’dae. L. gen. fem. n. merdae, of faeces)	
A bacterial species identified by metagenomic analyses. This species includes all bacteria with genomes that show ≥95% average nucleotide identity (ANI) to the type genome for the species to which we have assigned the MAG ID E3_87 and which is available via NCBI BioSample SAMN18472489. The GC content of the type genome is 52.7% and the genome length is 1.11 Mbp.	
Description of Candidatus Hippobium gen. nov.	
Candidatus Hippobium (Hip.po’bi.um. Gr. masc./fem. n. hippos, a horse; Gr. masc. n. bios, life; N.L. neut. n. Hippobium, a microbe associated with horses)	
A bacterial genus identified by metagenomic analyses. The genus includes all bacteria with genomes that show ≥60% average amino acid identity (AAI) to the genome of the type strain from the type species Candidatus Hippobium faecium. This genus has been assigned by GTDB-Tk v1.3.0 working on GTDB Release 06-RS202 (Olm et al., 2017; Scales et al., 2018) to the order UBA5829 and to the family UBA5829.	
Description of Candidatus Hippobium faecium sp. nov.	
Candidatus Hippobium faecium (fae’ci.um. L. fem. n. faex, dregs; L. gen. pl. n. faecium, of the dregs, of faeces)	
A bacterial species identified by metagenomic analyses. This species includes all bacteria with genomes that show ≥95% average nucleotide identity (ANI) to the type genome for the species to which we have assigned the MAG ID E3_206 and which is available via NCBI BioSample SAMN18472485. The GC content of the type genome is 39.1% and the genome length is 2.12 Mbp.	
Description of Candidatus Kurthia equi sp. nov.	
Candidatus Kurthia equi (e’qui. L. gen. masc. n. equi, of a horse)	
A bacterial species identified by metagenomic analyses. This species includes all bacteria with genomes that show ≥95% average nucleotide identity (ANI) to the type genome for the species to which we have assigned the MAG ID E1_MB2_88 and which is available via NCBI BioSample SAMN18472468. The GC content of the type genome is 35.7% and the genome length is 3.58 Mbp.	
Description of Candidatus Limimonas gen. nov.	
Candidatus Limimonas (Li.mi.mo’nas. L. masc. n. limus, dung; L. fem. n. monas, a monad; N.L. fem. n. Limimonas, a microbe associated with faeces)	
A bacterial genus identified by metagenomic analyses. The genus includes all bacteria with genomes that show ≥60% average amino acid identity (AAI) to the genome of the type strain from the type species Candidatus Limimonas coprohippi. This is a new name for the GTDB alphanumeric genus UBA1227, which is found in diverse mammalian guts. This genus has been assigned by GTDB-Tk v1.3.0 working on GTDB Release 06-RS202 (Olm et al., 2017; Scales et al., 2018) to the order Oscillospirales and to the family Acutalibacteraceae.	
Description of Candidatus Limimonas coprohippi sp. nov.	
Candidatus Limimonas coprohippi (co.pro.hip’pi. Gr. fem. n. kopros, dung; Gr. masc./fem. n. hippos, a horse; N.L. gen. n. coprohippi, associated with the faeces of horses)	
A bacterial species identified by metagenomic analyses. This species includes all bacteria with genomes that show ≥95% average nucleotide identity (ANI) to the type genome for the species to which we have assigned the MAG ID E1_MB2_82 and which is available via NCBI BioSample SAMN18472467. The GC content of the type genome is 40.5% and the genome length is 1.33 Mbp.	
Description of Candidatus Limimonas egerieequi sp. nov.	
Candidatus Limimonas egerieequi (e.ge.ri.e.e’qui. L. fem. n. egeries, dung; L. masc. n. equus, a horse; N.L. gen. n. egerieequi, associated with the faeces of horses)	
A bacterial species identified by metagenomic analyses. This species includes all bacteria with genomes that show ≥95% average nucleotide identity (ANI) to the type genome for the species to which we have assigned the MAG ID E5_MB2_129 and which is available via NCBI BioSample SAMN18472544. The GC content of the type genome is 41.8% and the genome length is 1.70 Mbp.	
Description of Candidatus Limimorpha caballi sp. nov.	
Candidatus Limimorpha caballi (ca.bal’li. L. gen. masc. n. caballi, of a horse)	
A bacterial species identified by metagenomic analyses. This species includes all bacteria with genomes that show ≥95% average nucleotide identity (ANI) to the type genome for the species to which we have assigned the MAG ID E5_119 and which is available via NCBI BioSample SAMN18472534. The GC content of the type genome is 48.3% and the genome length is 2.76 Mbp.	
Description of Candidatus Limimorpha equi sp. nov.	
Candidatus Limimorpha equi (e’qui. L. gen. masc. n. equi, of a horse)	
A bacterial species identified by metagenomic analyses. This species includes all bacteria with genomes that show ≥95% average nucleotide identity (ANI) to the type genome for the species to which we have assigned the MAG ID E1_MB2_99 and which is available via NCBI BioSample SAMN18472470. The GC content of the type genome is 45.1% and the genome length is 2.72 Mbp.	
Description of Candidatus Liminaster gen. nov.	
Candidatus Liminaster (Li.mi.nas’ter. L. masc. n. limus, dung; Gr. masc. n. naster, an inhabitant; N.L. masc. n. Liminaster, a microbe associated with faeces)	
A bacterial genus identified by metagenomic analyses. The genus includes all bacteria with genomes that show ≥60% average amino acid identity (AAI) to the genome of the type strain from the type species Candidatus Liminaster caballi. This is a new name for the GTDB alphanumeric genus UBA3663, which is found in diverse mammalian guts. This genus has been assigned by GTDB-Tk v1.3.0 working on GTDB Release 06-RS202 (Olm et al., 2017; Scales et al., 2018) to the order Bacteroidales and to the family UBA3663.	
Description of Candidatus Liminaster caballi sp. nov.	
Candidatus Liminaster caballi (ca.bal’li. L. gen. masc. n. caballi, of a horse)	
A bacterial species identified by metagenomic analyses. This species includes all bacteria with genomes that show ≥95% average nucleotide identity (ANI) to the type genome for the species to which we have assigned the MAG ID E4_95 and which is available via NCBI BioSample SAMN18472511. The GC content of the type genome is 50.1% and the genome length is 2.94 Mbp.	
Description of Candidatus Merdinaster gen. nov.	
Candidatus Merdinaster (Mer.di.nas’ter. L. fem. n. merda, dung; Gr. masc. n. naster, an inhabitant; N.L. masc. n. Merdinaster, a microbe associated with faeces)	
A bacterial genus identified by metagenomic analyses. The genus includes all bacteria with genomes that show ≥60% average amino acid identity (AAI) to the genome of the type strain from the type species Candidatus Merdinaster equi. This is a new name for the GTDB alphanumeric genus UBA7050, which is found in diverse mammalian guts. This genus has been assigned by GTDB-Tk v1.3.0 working on GTDB Release 06-RS202 (Olm et al., 2017; Scales et al., 2018) to the order Lachnospirales and to the family Lachnospiraceae,	
Description of Candidatus Merdinaster equi sp. nov.	
Candidatus Merdinaster equi (e’qui. L. gen. masc. n. equi, of a horse)	
A bacterial species identified by metagenomic analyses. This species includes all bacteria with genomes that show ≥95% average nucleotide identity (ANI) to the type genome for the species to which we have assigned the MAG ID E4_MB2_128 and which is available via NCBI BioSample SAMN18472516. The GC content of the type genome is 40.7% and the genome length is 1.95 Mbp.	
Description of Candidatus Methanocorpusculum equi sp. nov.	
Candidatus Methanocorpusculum equi (e’qui. L. gen. masc. n. equi, of a horse)	
A bacterial species identified by metagenomic analyses. This species includes all bacteria with genomes that show ≥95% average nucleotide identity (ANI) to the type genome for the species to which we have assigned the MAG ID E2_MB2_79 and which is available via NCBI BioSample SAMN18472479. The GC content of the type genome is 50.2% and the genome length is 1.15 Mbp.	
Description of Candidatus Minthenecus gen. nov.	
Candidatus Minthenecus (Minth.en.e’cus. Gr. masc. n. minthos, dung; N.L. masc. n. enecus, an inhabitant; N.L. masc. n. Minthenecus, a microbe associated with faeces)	
A bacterial genus identified by metagenomic analyses. The genus includes all bacteria with genomes that show ≥60% average amino acid identity (AAI) to the genome of the type strain from the type species Candidatus Minthenecus merdequi. This is a new name for the GTDB alphanumeric genus SFVR01, which is found in diverse mammalian guts. This genus has been assigned by GTDB-Tk v1.3.0 working on GTDB Release 06-RS202 (Olm et al., 2017; Scales et al., 2018) to the order Bacteroidales and to the family Paludibacteraceae.	
Description of Candidatus Minthenecus merdequi sp. nov.	
Candidatus Minthenecus merdequi (merd.e’qui. L. fem. n. merda, faeces; L. masc. n. equus, a horse; N.L. gen. n. merdequi, associated with the faeces of horses)	
A bacterial species identified by metagenomic analyses. This species includes all bacteria with genomes that show ≥95% average nucleotide identity (ANI) to the type genome for the species to which we have assigned the MAG ID E5_MB2_18 and which is available via NCBI BioSample SAMN18472545. The GC content of the type genome is 42.5% and the genome length is 1.80 Mbp.	
Description of Candidatus Minthocola gen. nov.	
Candidatus Minthocola (Min.tho’co.la. Gr. masc. n. minthos, dung; N.L. masc./fem. suffix -cola, an inhabitant; N.L. fem. n. Minthocola, a microbe associated with faeces)	
A bacterial genus identified by metagenomic analyses. The genus includes all bacteria with genomes that show ≥60% average amino acid identity (AAI) to the genome of the type strain from the type species Candidatus Minthocola equi. This is a new name for the GTDB alphanumeric genus UBA3774, which is found in diverse mammalian guts. This genus has been assigned by GTDB-Tk v1.3.0 working on GTDB Release 06-RS202 (Olm et al., 2017; Scales et al., 2018) to the order Lachnospirales and to the family Lachnospiraceae.	
Description of Candidatus Minthocola equi sp. nov.	
Candidatus Minthocola equi (e’qui. L. gen. masc. n. equi, of a horse)	
A bacterial species identified by metagenomic analyses. This species includes all bacteria with genomes that show ≥95% average nucleotide identity (ANI) to the type genome for the species to which we have assigned the MAG ID E5_MB2_38 and which is available via NCBI BioSample SAMN18472548. The GC content of the type genome is 45.2% and the genome length is 1.20 Mbp.	
Description of Candidatus Minthomonas gen. nov.	
Candidatus Minthomonas (Min.tho.mo’nas. Gr. masc. n. minthos, dung; L. fem. n. monas, a monad; N.L. fem. n. Minthomonas, a microbe associated with faeces)	
A bacterial genus identified by metagenomic analyses. The genus includes all bacteria with genomes that show ≥60% average amino acid identity (AAI) to the genome of the type strain from the type species Candidatus Minthomonas equi. This is a new name for the GTDB alphanumeric genus CAG-831, which is found in diverse mammalian guts. This genus has been assigned by GTDB-Tk v1.3.0 working on GTDB Release 06-RS202 (Olm et al., 2017; Scales et al., 2018) to the order Bacteroidales and to the family UBA932.	
Description of Candidatus Minthomonas equi sp. nov.	
Candidatus Minthomonas equi (e’qui. L. gen. masc. n. equi, of a horse)	
A bacterial species identified by metagenomic analyses. This species includes all bacteria with genomes that show ≥95% average nucleotide identity (ANI) to the type genome for the species to which we have assigned the MAG ID E5_18 and which is available via NCBI BioSample SAMN18472536. The GC content of the type genome is 47.6% and the genome length is 1.36 Mbp.	
Description of Candidatus Minthosoma gen. nov.	
Candidatus Minthosoma (Min.tho.so’ma. Gr. masc. n. minthos, dung; Gr. neut. n. soma, a body; N.L. neut. n. Minthosoma, a microbe associated with faeces)	
A bacterial genus identified by metagenomic analyses. The genus includes all bacteria with genomes that show ≥60% average amino acid identity (AAI) to the genome of the type strain from the type species Candidatus Minthosoma caballi. This is a new name for the GTDB alphanumeric genus UBA4334, which is found in diverse mammalian guts. This genus has been assigned by GTDB-Tk v1.3.0 working on GTDB Release 06-RS202 (Olm et al., 2017; Scales et al., 2018) to the order Bacteroidales and to the family Bacteroidaceae	
Description of Candidatus Minthosoma caballi sp. nov.	
Candidatus Minthosoma caballi (ca.bal’li. L. gen. masc. n. caballi, of a horse)	
A bacterial species identified by metagenomic analyses. This species includes all bacteria with genomes that show ≥95% average nucleotide identity (ANI) to the type genome for the species to which we have assigned the MAG ID E5_9 and which is available via NCBI BioSample SAMN18472539. The GC content of the type genome is 44.2% and the genome length is 3.21 Mbp.	
Description of Candidatus Minthosoma equi sp. nov.	
Candidatus Minthosoma equi (e’qui. L. gen. masc. n. equi, of a horse)	
A bacterial species identified by metagenomic analyses. This species includes all bacteria with genomes that show ≥95% average nucleotide identity (ANI) to the type genome for the species to which we have assigned the MAG ID E4_MB2_18 and which is available via NCBI BioSample SAMN18472519. The GC content of the type genome is 44.1% and the genome length is 3.51 Mbp.	
Description of Candidatus Minthousia gen. nov.	
Candidatus Minthousia (Minth.ou’s.ia. Gr. masc. n. minthos, dung; Gr. fem. n. ousia, an essence; N.L. fem. n. Minthousia, a microbe associated with faeces)	
A bacterial genus identified by metagenomic analyses. The genus includes all bacteria with genomes that show ≥60% average amino acid identity (AAI) to the genome of the type strain from the type species Candidatus Minthousia equi. This is a new name for the GTDB alphanumeric genus UBA4293, which is found in diverse mammalian guts. This genus has been assigned by GTDB-Tk v1.3.0 working on GTDB Release 06-RS202 (Olm et al., 2017; Scales et al., 2018) to the order Bacteroidales and to the family Bacteroidaceae.	
Description of Candidatus Minthousia equi sp. nov.	
Candidatus Minthousia equi (e’qui. L. gen. masc. n. equi, of a horse)	
A bacterial species identified by metagenomic analyses. This species includes all bacteria with genomes that show ≥95% average nucleotide identity (ANI) to the type genome for the species to which we have assigned the MAG ID E4_55 and which is available via NCBI BioSample SAMN18472509. The GC content of the type genome is 42.9% and the genome length is 2.61 Mbp.	
Description of Candidatus Neoflavobacterium gen. nov.	
Candidatus Neoflavobacterium gen. nov.	
(Ne.o.fla.vo.bac.te.ri.um Gr. masc. adj. νέος new N.L. neut. n. Flavobacterium An existing genus Neoflavobacterium N.L. neut. n. A genus related to but distinct from the existing genus Flavobacterium)	
A bacterial genus identified by metagenomic analyses. The genus includes all bacteria with genomes that show ≥60% average amino acid identity (AAI) to the genome of the type strain from the type species Candidatus Neoflavobacterium equi. This genus has been assigned by GTDB-Tk v1.3.0 working on GTDB Release 06-RS202 (Olm et al., 2017; Scales et al., 2018) to the order Flavobacteriales and to the family Flavobacteriaceae	
Description of Candidatus Neoflavobacterium equi sp. nov.	
Candidatus Neoflavobacterium equi sp. nov. (e’qui. L. gen. masc. n. equi, of a horse)	
A bacterial species identified by metagenomic analyses. This species includes all bacteria with genomes that show ≥95% average nucleotide identity (ANI) to the type genome for the species to which we have assigned the MAG ID E2_MB2_6 and which is available via NCBI BioSample SAMN18472477. The GC content of the type genome is 37.7% and the genome length is 2.17 Mbp.	
Description of Candidatus Onthonaster gen. nov.	
Candidatus Onthonaster (On.tho.nas’ter. Gr. masc. n. onthos, dung; Gr. masc. n. naster, an inhabitant; N.L. masc. n. Onthonaster, a microbe associated with faeces)	
A bacterial genus identified by metagenomic analyses. The genus includes all bacteria with genomes that show ≥60% average amino acid identity (AAI) to the genome of the type strain from the type species Candidatus Onthonaster equi. This is a new name for the GTDB alphanumeric genus YIM-102668, which is found in diverse mammalian guts. This genus has been assigned by GTDB-Tk v1.3.0 working on GTDB Release 06-RS202 (Olm et al., 2017; Scales et al., 2018) to the order Flavobacteriales and to the family Weeksellaceae	
Description of Candidatus Onthonaster equi sp. nov.	
Candidatus Onthonaster equi (e’qui. L. gen. masc. n. equi, of a horse)	
A bacterial species identified by metagenomic analyses. This species includes all bacteria with genomes that show ≥95% average nucleotide identity (ANI) to the type genome for the species to which we have assigned the MAG ID E1_98 and which is available via NCBI BioSample SAMN18472458. This is a new name for the alphanumeric GTDB species sp003687725, which is found in diverse mammalian guts. The GC content of the type genome is 31.1% and the genome length is 2.30 Mbp.	
Description of Candidatus Phascolarctobacterium caballi sp. nov.	
Candidatus Phascolarctobacterium caballi (ca.bal’li. L. gen. masc. n. caballi, of a horse)	
A bacterial species identified by metagenomic analyses. This species includes all bacteria with genomes that show ≥95% average nucleotide identity (ANI) to the type genome for the species to which we have assigned the MAG ID E4_135 and which is available via NCBI BioSample SAMN18472503. The GC content of the type genome is 39.4% and the genome length is 1.56 Mbp.	
Description of Candidatus Phascolarctobacterium equi sp. nov.	
Candidatus Phascolarctobacterium equi (e’qui. L. gen. masc. n. equi, of a horse)	
A bacterial species identified by metagenomic analyses. This species includes all bacteria with genomes that show ≥95% average nucleotide identity (ANI) to the type genome for the species to which we have assigned the MAG ID E2_44 and which is available via NCBI BioSample SAMN18472473. The GC content of the type genome is 46.7% and the genome length is 0.93 Mbp.	
Description of Candidatus Physcocola gen. nov.	
Candidatus Physcocola (Phys.co’co.la. Gr. fem. n. physke, the colon; N.L. masc./fem. suffix –cola, an inhabitant; N.L. fem. n. Physcocola, a microbe associated with the large intestine)	
A bacterial genus identified by metagenomic analyses. The genus includes all bacteria with genomes that show ≥60% average amino acid identity (AAI) to the genome of the type strain from the type species Candidatus Physcocola equi. This is a new name for the GTDB alphanumeric genus UBA4345, which is found in diverse mammalian guts. This genus has been assigned by GTDB-Tk v1.3.0 working on GTDB Release 06-RS202 (Olm et al., 2017; Scales et al., 2018) to the order Bacteroidales and to the family Paludibacteraceae.	
Description of Candidatus Physcocola equi sp. nov.	
Candidatus Physcocola equi (e’qui. L. gen. masc. n. equi, of a horse)	
A bacterial species identified by metagenomic analyses. This species includes all bacteria with genomes that show ≥95% average nucleotide identity (ANI) to the type genome for the species to which we have assigned the MAG ID E4_MB2_42 and which is available via NCBI BioSample SAMN18472523. The GC content of the type genome is 43.3% and the genome length is 2.99 Mbp.	
Description of Candidatus Physcosoma gen. nov.	
Candidatus Physcosoma (Phys.co.so’ma. Gr. fem. n. physke, the colon; Gr. neut. n. soma, a body; N.L. neut. n. Physcosoma, a microbe associated with the large intestine)	
A bacterial genus identified by metagenomic analyses. The genus includes all bacteria with genomes that show ≥60% average amino acid identity (AAI) to the genome of the type strain from the type species Candidatus Physcosoma equi. This is a new name for the GTDB alphanumeric genus UBA5920, which is found in diverse mammalian guts. This genus has been assigned by GTDB-Tk v1.3.0 working on GTDB Release 06-RS202 (Olm et al., 2017; Scales et al., 2018) to the order Sphaerochaetales and to the family Sphaerochaetaceae.	
Description of Candidatus Physcosoma equi sp. nov.	
Candidatus Physcosoma equi (e’qui. L. gen. masc. n. equi, of a horse)	
A bacterial species identified by metagenomic analyses. This species includes all bacteria with genomes that show ≥95% average nucleotide identity (ANI) to the type genome for the species to which we have assigned the MAG ID E4_160 and which is available via NCBI BioSample SAMN18472504. The GC content of the type genome is 49.1% and the genome length is 2.06 Mbp.	
Description of Candidatus Physcousia gen. nov.	
Candidatus Physcousia (Physc.ou’si.a. Gr. fem. n. physke the colon; Gr. fem. n. ousia, an essence.e; N.L. fem. n. Physcousia, a microbe associated with the large intestine)	
A bacterial genus identified by metagenomic analyses. The genus includes all bacteria with genomes that show ≥60% average amino acid identity (AAI) to the genome of the type strain from the type species Candidatus Physcousia caballi. This is a new name for the GTDB alphanumeric genus UBA4372, which is found in diverse mammalian guts. This genus has been assigned by GTDB-Tk v1.3.0 working on GTDB Release 06-RS202 (Olm et al., 2017; Scales et al., 2018) to the order Bacteroidales and to the family Bacteroidaceae.	
Description of Candidatus Physcousia caballi sp. nov.	
Candidatus Physcousia caballi (ca.bal’li. L. gen. masc. n. caballi, of a horse)	
A bacterial species identified by metagenomic analyses. This species includes all bacteria with genomes that show ≥95% average nucleotide identity (ANI) to the type genome for the species to which we have assigned the MAG ID E4_MB2_73 and which is available via NCBI BioSample SAMN18472529. The GC content of the type genome is 50.5% and the genome length is 3.81 Mbp.	
Description of Candidatus Physcousia equi sp. nov.	
Candidatus Physcousia equi (e’qui. L. gen. masc. n. equi, of a horse)	
A bacterial species identified by metagenomic analyses. This species includes all bacteria with genomes that show ≥95% average nucleotide identity (ANI) to the type genome for the species to which we have assigned the MAG ID E4_MB2_112 and which is available via NCBI BioSample SAMN18472514. The GC content of the type genome is 52.4% and the genome length is 2.43 Mbp.	
Description of Candidatus Prevotella equi sp. nov.	
Candidatus Prevotella equi (e’qui. L. gen. masc. n. equi, of a horse)	
A bacterial species identified by metagenomic analyses. This species includes all bacteria with genomes that show ≥95% average nucleotide identity (ANI) to the type genome for the species to which we have assigned the MAG ID E4_23 and which is available via NCBI BioSample SAMN18472507. The GC content of the type genome is 44.5% and the genome length is 3.45 Mbp.	
Description of Candidatus Ruminococcus equi sp. nov.	
Candidatus Ruminococcus equi (e’qui. L. gen. masc. n. equi, of a horse)	
A bacterial species identified by metagenomic analyses. This species includes all bacteria with genomes that show ≥95% average nucleotide identity (ANI) to the type genome for the species to which we have assigned the MAG ID E3_41 and which is available via NCBI BioSample SAMN18472487. GTDB has assigned this species to a genus marked with an alphabetical suffix. However, as this genus designation cannot be incorporated into a well-formed binomial, in naming. this species, we have used the current validly published name for the genus. The GC content of the type genome is 39.9% and the genome length is 1.73 Mbp.	
Description of Candidatus Scatonaster gen. nov.	
Candidatus Scatonaster (Sca.to.nas’ter. Gr. neut. n. skor, skatos, dung; Gr. masc. n. naster, an inhabitant; N.L. masc. n. Scatonaster a microbe associated with faeces)	
A bacterial genus identified by metagenomic analyses. The genus includes all bacteria with genomes that show ≥60% average amino acid identity (AAI) to the genome of the type strain from the type species Candidatus Scatonaster coprocaballi. This is a new name for the GTDB alphanumeric genus Firm-16, which is found in diverse mammalian guts. This genus has been assigned by GTDB-Tk v1.3.0 working on GTDB Release 06-RS202 (Olm et al., 2017; Scales et al., 2018) to the order Saccharofermentanales and to the family Saccharofermentanaceae.	
Description of Candidatus Scatonaster coprocaballi sp. nov.	
Candidatus Scatonaster coprocaballi (co.pro.ca.bal’li. Gr. fem. n. kopros, dung; L. masc. n. caballus, a horse; N.L. gen. n. coprocaballi, associated with the faeces of horses)	
A bacterial species identified by metagenomic analyses. This species includes all bacteria with genomes that show ≥95% average nucleotide identity (ANI) to the type genome for the species to which we have assigned the MAG ID E5_MB2_10 and which is available via NCBI BioSample SAMN18472540. The GC content of the type genome is 46.9% and the genome length is 2.23 Mbp.	
Description of Candidatus Scybalocola gen. nov.	
Candidatus Scybalocola (Scy.ba.lo’co.la. Gr. neut. n. skybalon, dung; N.L. masc./fem. suffix –cola, an inhabitant; N.L. fem. n. Scybalocola a microbe associated with faeces)	
A bacterial genus identified by metagenomic analyses. The genus includes all bacteria with genomes that show ≥60% average amino acid identity (AAI) to the genome of the type strain from the type species Candidatus Scybalocola fimicaballi. This is a new name for the GTDB alphanumeric genus UBA1723, which is found in diverse mammalian guts. This genus has been assigned by GTDB-Tk v1.3.0 working on GTDB Release 06-RS202 (Olm et al., 2017; Scales et al., 2018) to the order Bacteroidales and to the family Paludibacteraceae.	
Description of Candidatus Scybalocola fimicaballi sp. nov.	
Candidatus Scybalocola fimicaballi (fi.mi.ca.bal’li. L. masc. n. fimus, dung; L. masc. n. caballus, a horse; N.L. gen. n. fimicaballi, associated with the faeces of horses)	
A bacterial species identified by metagenomic analyses. This species includes all bacteria with genomes that show ≥95% average nucleotide identity (ANI) to the type genome for the species to which we have assigned the MAG ID E1_25 and which is available via NCBI BioSample SAMN18472456. This is a new name for the alphanumeric GTDB species sp002317115, which is found in diverse mammalian guts. The GC content of the type genome is 41.7% and the genome length is 3.11 Mbp.	
Description of Candidatus Scybalousia gen. nov.	
Candidatus Scybalousia (Scy.bal.ou’s.ia. Gr. neut. n. skybalon, dung; Gr. fem. n. ousia, an essence; N.L. fem n. Scybalousia, a microbe associated with faeces)	
A bacterial genus identified by metagenomic analyses. The genus includes all bacteria with genomes that show ≥60% average amino acid identity (AAI) to the genome of the type strain from the type species Candidatus Scybalousia scubalohippi. This is a new name for the GTDB alphanumeric genus Phil12, which is found in diverse mammalian guts. This genus has been assigned by GTDB-Tk v1.3.0 working on GTDB Release 06-RS202 (Olm et al., 2017; Scales et al., 2018) to the order Bacteroidales and to the family P3.	
Description of Candidatus Scybalousia scybalohippi sp. nov.	
Candidatus Scybalousia scybalohippi (scy.ba.lo.hip’pi. Gr. neut. n. skybalon, dung; Gr. masc./fem. n. hippos, a horse; N.L. gen. n. scybalohippi, associated with the faeces of horses)	
A bacterial species identified by metagenomic analyses. This species includes all bacteria with genomes that show ≥95% average nucleotide identity (ANI) to the type genome for the species to which we have assigned the MAG ID E3_144 and which is available via NCBI BioSample SAMN18472482. The GC content of the type genome is 35.4% and the genome length is 2.63 Mbp.	
Description of Candidatus Sodaliphilus aphodohippi sp. nov.	
Candidatus Sodaliphilus aphodohippi (aph.o.do.hip’pi. Gr. fem. n. aphodos, dung; Gr. masc./fem. n. hippos, a horse; N.L. gen. n. aphodohippi, associated with the faeces of horses)	
A bacterial species identified by metagenomic analyses. This species includes all bacteria with genomes that show ≥95% average nucleotide identity (ANI) to the type genome for the species to which we have assigned the MAG ID E3_0 and which is available via NCBI BioSample SAMN18472480. The GC content of the type genome is 50% and the genome length is 2.49 Mbp.	
Description of Candidatus Sodaliphilus fimicaballi sp. nov.	
Candidatus Sodaliphilus fimicaballi (fi.mi.ca.bal’li. L. masc. n. fimus, dung; L. masc. n. caballus, a horse; N.L. gen. n. fimicaballi, associated with the faeces of horses)	
A bacterial species identified by metagenomic analyses. This species includes all bacteria with genomes that show ≥95% average nucleotide identity (ANI) to the type genome for the species to which we have assigned the MAG ID E4_193 and which is available via NCBI BioSample SAMN18472506. The GC content of the type genome is 48.1% and the genome length is 2.46 Mbp.	
Description of Candidatus Sodaliphilus limicaballi sp. nov.	
Candidatus Sodaliphilus limicaballi (li.mi.ca.bal’li. L. masc. n. limus, dung; L. masc. n. caballus, a horse; N.L. gen. n. limicaballi, associated with the faeces of horses)	
A bacterial species identified by metagenomic analyses. This species includes all bacteria with genomes that show ≥95% average nucleotide identity (ANI) to the type genome for the species to which we have assigned the MAG ID E2_8 and which is available via NCBI BioSample SAMN18472474. The GC content of the type genome is 50.4% and the genome length is 3.16 Mbp.	
Description of Candidatus Treponema caballi sp. nov.	
Candidatus Treponema caballi (ca.bal’li. L. gen. masc. n. caballi, of a horse)	
A bacterial species identified by metagenomic analyses. This species includes all bacteria with genomes that show ≥95% average nucleotide identity (ANI) to the type genome for the species to which we have assigned the MAG ID E1_106 and which is available via NCBI BioSample SAMN18472451. GTDB has assigned this species to a genus marked with an alphabetical suffix. However, as this genus designation cannot be incorporated into a well-formed binomial, in naming. this species, we have used the current validly published name for the genus. The GC content of the type genome is 47.1% and the genome length is 2.91 Mbp.	
Description of Candidatus Treponema equi sp. nov.	
Candidatus Treponema equi (e’qui. L. gen. masc. n. equi, of a horse)	
A bacterial species identified by metagenomic analyses. This species includes all bacteria with genomes that show ≥95% average nucleotide identity (ANI) to the type genome for the species to which we have assigned the MAG ID E4_MB2_46 and which is available via NCBI BioSample SAMN18472525. GTDB has assigned this species to a genus marked with an alphabetical suffix. However, as this genus designation cannot be incorporated into a well-formed binomial, in naming. this species, we have used the current validly published name for the genus. The GC content of the type genome is 44.3% and the genome length is 1.79 Mbp.	
Description of Candidatus Treponema equifaecale sp. nov.	
Candidatus Treponema equifaecale (e.qui.fae.ca’le. L. masc. n. equus, a horse; N.L. neut. adj. faecale, faecal; N.L. neut. adj. equifaecale, associated with the faeces of horses)	
A bacterial species identified by metagenomic analyses. This species includes all bacteria with genomes that show ≥95% average nucleotide identity (ANI) to the type genome for the species to which we have assigned the MAG ID E4_MB2_2 and which is available via NCBI BioSample SAMN18472520. GTDB has assigned this species to a genus marked with an alphabetical suffix. However, as this genus designation cannot be incorporated into a well-formed binomial, in naming. this species, we have used the current validly published name for the genus. The GC content of the type genome is 40.2% and the genome length is 2.81 Mbp.	
Description of Candidatus Treponema merdequi sp. nov.	
Candidatus Treponema merdequi (merd.e’qui. L. fem. n. merda, faeces; L. masc. n. equus, a horse; N.L. gen. n. merdequi, associated with the faeces of horses)	
A bacterial species identified by metagenomic analyses. This species includes all bacteria with genomes that show ≥95% average nucleotide identity (ANI) to the type genome for the species to which we have assigned the MAG ID E5_50 and which is available via NCBI BioSample SAMN18472538. GTDB has assigned this species to a genus marked with an alphabetical suffix. However, as this genus designation cannot be incorporated into a well-formed binomial, in naming. this species, we have used the current validly published name for the genus. The GC content of the type genome is 35.8% and the genome length is 2.70 Mbp.	
Description of Candidatus Treponema scatequi sp. nov.	
Candidatus Treponema scatequi (scat.e’qui. Gr. neut. n. skor, skatos, dung; L. masc. n. equus, a horse; N.L. gen. n. scatequi, associated with the faeces of horses)	
A bacterial species identified by metagenomic analyses. This species includes all bacteria with genomes that show ≥95% average nucleotide identity (ANI) to the type genome for the species to which we have assigned the MAG ID E1_MB2_111 and which is available via NCBI BioSample SAMN18472460. GTDB has assigned this species to a genus marked with an alphabetical suffix. However, as this genus designation cannot be incorporated into a well-formed binomial, in naming this species, we have used the current validly published name for the genus. The GC content of the type genome is 38.4% and the genome length is 2.31 Mbp.	

A novel class within the Armatimonadetes

One of our MAGs—and the associated species cluster, which we have called Ca. Hippobium faecium—was assigned to the family of alphanumeric designation UBA5829. Assigned by GTDB to its own class, order and family, all members of this family belong to the recently named phylum Armatimonadetes (also called Armatimonadota; previously known as OP10) (Tamaki et al., 2011). Scrutiny of the NCBI database in August 2021 reveals that no genome assemblies linked to this phylum originate from the vertebrate gut, instead being metagenome-assembled genomes largely derived from bioreactors. Ca. Hippobium faecium was found at >1× coverage in two samples (SAMN13344080 & SAMN13344082), with relative abundance of this species across both samples being 94% and 4% respectively.

Distribution and metabolism

Our de-replicated high- and medium-quality MAGs account for 18% (±5%) of our host-depleted metagenomic reads. Distribution analysis identified 17 species present at ≥1× coverage in all samples (core MAGs represent 15% of our dereplicated MAG catalogue), spanning four bacterial phyla and the archaea (Fig. 2A and Table S7). No species were present at ≥10× coverage in all samples. While the majority of identified MAG species clusters had predominant relative abundance in only one sample, species including Ca. Methanocorpusculum equi, Acinetobacter lanii, Ca. Colimonas fimequi and Ca. Colisoma equi had more uniform distribution across all sample indicating a more central function in equine health. Species quantification shows a steady incline in the cumulative number of species identified when successively adding each of the five separate horse faecal samples (Fig. 2B). While species in the genus Acinetobacter represented seven of the ten most abundant species in our samples according to the Kraken2 analysis, only one Acinetobacter species was identified within out MAG catalogue. It seems likely that the presence of multiple closely related strains and species from this genus increases the likelihood of chimeric, unresolved bins. More generally, high abundance of a genus in the Kraken2/bracken analysis is not strongly linked to the recovery of medium or high quality MAG from that genus.

Figure 2 Distribution and metabolism of equine microbial genomes.

(A). Heat map depicting the relative abundance of 110 taxa across five metagenomic samples. MAG species clusters have been annotated with their taxonomic class and species assignment. All data were Log10 transformed with Blue colour depicting species of low abundance and Red showing high abundance. (B) Species accumulation curve based on species count data for 110 identified MAG species clusters over five metagenomic samples. Species accumulation curve has been created using the Specaccum function of the R vegan package. (C) Percentage of annotated CAZyme genes per taxa assigned to different CAZyme functional classes. Where multiple MAGs were assigned to the same species cluster (95% ANI), an average annotation percentage was calculated. Species have been ordered according to GTDB-tk assigned phylum. Functional classes are depicted by bar colour; Auxiliary Activities (AA), Carbohydrate-Binding Modules (CBM), Carbohydrate Esterases (CE), Glycoside Hydrolases (GH), GlycosylTransferases (GT), Polysaccharide Lysases (PL).

We created a catalogue of 228,125 genes from our medium- and high-quality MAGs. All 123 MAGs encoded known carbohydrate-active enzymes (CAZymes), with an average of 69 CAZymes per genome (Table S8). Most (>70%) MAG species clusters with a higher-than average repertoire of CAZymes belonged to the Bacteroidota. Of the ~8,500 CAZyme genes reported, most were associated with classes devoted to assembly (glycosyltransferases [GT] 29%) and breakdown (glycoside hydrolases [GH] 51%) of carbohydrate complexes, with far fewer from other groups of CAZymes; being the polysaccharide lyases (PL) and carbohydrate esterases (CE) alongside two further non-enzymatic groups being the carbohydrate-binding modules (CBM) and the auxiliary activities (AA). (Fig. 2C). Recovery of 93 classes of glycoside hydrolases from the equine gut mirrors similar enzymes in the sheep rumen linked to fibre degradation (He et al., 2019). Over half of our equine MAGs encode CAZymes with presumed involvement in degradation of hemi-cellulose (58%), cellulose (51%) or pectin or soluble fibre (>60%).

Many novel bacteriophage genomes

The program VirSorter classified 2,500 contigs as “highly likely” or “likely” to originate from bacteriophages (Table S9). Of these, 190 bacteriophage genomes were identified as “high-quality” (n = 181) or “complete” (n = 9) after de-replication (Fig. 3A). However, as none showed close identity to known viral sequences, they all represent novel bacteriophage species. Genome sizes ranged from 5 to 145 kb, including 42 genomes ranging from 5 to 15 kb in length. Using the viral taxonomy tool Demovir (https://github.com/feargalr/Demovir), we could assign 150 of these new phages to known viral families. An additional 29 could be assigned to taxonomically informative viral clusters, based on similarities between predicted proteins from our contigs and proteins from the viral component of the RefSeq94 database (Table S10). Just under half (n = 14) of these viral clusters contained at least one reference genome, thus expanding the known diversity of four viral families (Fig. 3B).

Figure 3 Bacteriophage analysis of equine faecal samples from five thoroughbred horses.

(A) CheckV quality tiers vs contig length (provided as Log10 values). (B) Protein sharing network of 190 High-quality or Complete phage genomes assembled from five equine faecal metagenomes and compared against a de-replicated RefSeq database of reference prokaryotic virus genomes. Each node represents a viral genome, with node colour depicting source sample and node size scales according to metagenome contig length. Grey nodes depict reference genomes, with no size scaling shown. Network edges indicate statistically significant relationships between the protein profiles of respective viral genomes. Annotation has been provided to highlight viral clusters of interest. (C) Upset plot of phage genomes shared between or specific to source faecal sample, set colour is defined by sample. Each bar represents the number of phage genomes described within the given samples.

Almost all of our viral genomes represented tailed dsDNA phages from the order Caudovirales (Babenko et al., 2020) and could be sub-classified into the families Siphoviridae (73%), Podoviridae or the newly delineated Schitoviridae (Wittmann et al., 2020) (21%) and Myoviridae (6%). Seven genomes were assigned to ssDNA viruses from the family Microviridae, four of which cluster as part of the subfamily Gokushovirinae. Weak connections of three viral genomes to a viral cluster of Obolenskvirus, whose known members all infect Acinetobacter sp., likely indicates the presence of novel bacteriophage genomes predating on the prominent population of bacterial Acinetobacter within the equine hind-gut. Present within the viral cluster network but notably absent within our bacteriophage catalogue included the model Escherichia coli phages T4 (Tevenvirinae) and T7 (Studiervirinae) or the Mycobacterium infecting actinophages. We observed several novel viral clusters comprising only genomes assembled in this study, which could be classified as the first representatives of new horse hindgut-associated phage families. Based on the proteome comparisons (Fig. 3B), we predict at least three new families.

Over three quarters of the recovered phage genomes were found at >1 × coverage in just a single sample (Fig. 3C), with observed phage genomes ranging in coverage from 27 to 65. Similar inter-individual variation in phage abundance and diversity has been described within the human gut microbiome despite evidence of strong temporal stability within individuals (Ogilvie & Jones, 2015). Variation in phage composition between samples is probably driven by environmental and host-derived factors, although the balance of these influences is yet to be defined (Duerkop, 2018). Only one phage was found in all five samples, with coverage ranging from 1.9×–29× and forming a cluster with Lactococcus phage P087 of the family Siphoviridae. The small sample size makes it impractical at this stage to define a core virome for the horse using criteria applied to the human gut microbiome (Manrique et al., 2016).

Discussion

Compared to the human gut, the microbiology of the horse gut remains largely unexplored. Here, we deliver new insights into this important ecosystem while also showcasing the advantages of shotgun metagenomics in providing catalogues of genes and genome sequences that take us well beyond what can be achieved using 16S ribosomal RNA gene sequences. Exploration of just five faecal samples allowed discovery of—and recovery of—genomes from nearly 100 new bacterial and archaeal species and nearly 200 bacteriophage genomes, substantially increasing the known microbial diversity of this environment. Deposition of genomes from these species into publicly available databases will underpin all future studies, improving the quality of reference-based taxonomic assignments.

While the limited scope of this study means it cannot hope to provide a comprehensive view of taxonomic diversity within the horse gut, it gives us a tantalizing glimpse of the richness that awaits us when such approaches are rolled out more widely, particularly as integration of long-read sequencing into metagenomics brings the promise of genome assemblies rivaling those from cultured isolates (Moss, Maghini & Bhatt, 2020; Nicholls et al., 2019). These advances will help to bridge the gap between the taxonomic profiles already defined through amplicon sequencing and newly uncovered MAGs by allowing incorporation of complex repetitive elements into assemblies, which are often missed by current assembly algorithms. Just as the horse allowed humans to explore new external landscapes, new sequencing and bioinformatics approaches will allow us to explore the inner world of the equine gut microbiome.

Conclusions

This research generates an introductory census of the thoroughbred horse gut microbiome and its associated metabolic potential far beyond the scope of that seen in currently available metagenomic studies, with these often relying upon 16S rRNA gene sequence analyses. Here, we present dozens of novel bacterial genera and species. Assignment of previously unnamed species to Candidatus binomials, as employed here, provides an important precedent for the continued description of these organism as they are uncovered in other biological environments.

Supplemental Information

Supplemental Information 1 Supporting information on all, phage contigs, raw read Metagenomic data and the resulting MAGs created as part of this project.

Table S1. Sequence Summaries. Summaries of sequencing data from 5 metagenomic samples sourced from equine faeces from BioProject PRJNA590977 Table S2. Read-based taxonomic analysis. Bracken read based relative abundance values for 5 equine faecal samples from BioProject PRJNA590977. Table S3. Metagenome Assembled Genome statistics. CheckM genome statistics for MAG catalogue following assembly of metagenomic reads from 5 horse faecal samples. For High and medium quality sequences, clusters at 95% and 99% ANI have been detailed.Table S4. tRNA presence in high and medium quality MAGs. tRNA presence across MAG catalogue following assembly of metagenomic reads from 5 horse faecal samples. Table S5. MAG taxonomic assignments. Taxonomic analysis for de-replicated MAG species clusters (95% ANI) according to GTDB (release 202), CAT/BAT (NCBI nr database) and ReferenceSeeker (RefSeq database). Newly assigned Latin binomials have been provided where appropriate. Table S6. AAI analysis for novel genera. Average amino acid identity (AAI) scores for all genomes of novel genera as determined by CompareM. Table S7. Distribution analysis of recovered MAGs. Coverage statistics for 110 MAG species clusters (95% ANI) recovered from 5 metagenomic samples derived from equine faeces. Relative abundance of all MAG species clusters across samples have been provided. Table S8. Functional annotation of recovered MAGs. Presence of genes associated with CAZyme function across 123 MAGs recovered from metagenomic reads of 5 horse faecal samples. Table S9. Genome and quality analysis of with recovered phage sequences. CheckV summary statistics of all VirSorter Category 1 and Category 2 phages >5 kb in length derived from metagenomic assemblies of horse faecal samples. For all High and medium quality phage sequences, further detail of taxonomic annotation and sequences coverage have been provided. Table S10. Protein based clustering of high quality or complete phage sequences. vCONTACT2 output for de-replicated catalogue of 190 phage genomes derived from equine faecal samples and classified as ‘High-quality’ or ‘Complete’ by CheckV.

Click here for additional data file.

The authors would like to thank all horse owners, stud farms and vets for facilitating the collection of horse faecal samples. The authors would also like to thank the core bioinformatics team working at Quadram Institute Bioscience for their help in data processing.

Additional Information and Declarations

Competing Interests

Author Contributions

Animal Ethics

Data Availability

The authors declare that they have no competing interests.

Rachel Gilroy performed the experiments, analyzed the data, prepared figures and/or tables, authored or reviewed drafts of the paper, and approved the final draft.

Joy Leng performed the experiments, authored or reviewed drafts of the paper, and approved the final draft.

Anuradha Ravi analyzed the data, authored or reviewed drafts of the paper, and approved the final draft.

Evelien M Adriaenssens analyzed the data, prepared figures and/or tables, authored or reviewed drafts of the paper, and approved the final draft.

Aharon Oren analyzed the data, authored or reviewed drafts of the paper, and approved the final draft.

Dave Baker performed the experiments, authored or reviewed drafts of the paper, and approved the final draft.

Roberto M La Ragione conceived and designed the experiments, authored or reviewed drafts of the paper, and approved the final draft.

Christopher Proudman conceived and designed the experiments, authored or reviewed drafts of the paper, and approved the final draft.

Mark J Pallen conceived and designed the experiments, prepared figures and/or tables, authored or reviewed drafts of the paper, and approved the final draft.

The following information was supplied relating to ethical approvals (i.e., approving body and any reference numbers):

Completed under The University of Surrey’s ethical review framework, project code: NERA-2017-007-SVM

The following information was supplied regarding data availability:

The data are available at BioProject: PRJNA590977 and Figshare: Gilroy, Rachel (2021): MAG catalogue - all. figshare. Dataset. https://doi.org/10.6084/m9.figshare.14268095.v1

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
