# Peer review of "Metagenomic investigation of the equine faecal microbiome reveals extensive taxonomic diversity"

_PeerJ, doi:10.7717/peerj.13084_

## Round 0.1 · original submission · Major Revisions

The manuscript has scope for publication but needs overwhelming revision. The authors have evaluated the taxonomic breakdown of the microbial community in the equine gut microbiome, the functional evaluation appears to be a preliminary.

·

Basic reporting

Dear editor and authors,

This is a novel and interesting study that adds to the current knowledge on the intestinal microbiota of horses. I have a few suggestions to improve this already good study.

Introduction gives too much emphasis on the importance of the horse, but lacks a deeper justification on the rationale and on the importance of the methods used in the current study. Authors should describe in more details the limitations of short reads sequencing and how the use of more comprehensive methods have aided the characterization of the microbiota of other environments such as soil and the human intestines.

While the discussion is extremely concise, many parts of the results are actually discussing and interpreting results. I recommend you to combine those two sessions.

Interesting aspects of the study deserve more attention, as the inter-horse variability, as evidenced in Figure 2.

>65% of reads were classified as unassigned. Most studies using V4 region find around 10% of unclassified bacteria that are frequently deemed to truly unknown organisms, but impossible to distinguish from sequencing errors. How likely your results are due to limitations of the methods used in this study?

Other limitations of the study need to be discussed. Maybe I am wrong, but I learned that sequencing based genome assemblies needed to be confirmed by culture based studies to prove the existence of those species. Isn’t necessary to culture a bacteria before you can definitively name it?

Experimental design

The study design is adequate to support the conclusions. I am not a bioinformatician, but data analysis seems to be performed by an expert in the field.

Validity of the findings

This is the first study using shotgun sequencing to analyze the equine microbiome and therefore is extremelly novel. Conclusions are supported by the results, but the article lacks further discussions about the possible limitations.

Additional comments

Other suggestions:
Line 86: please rephrase “reveal nothing about”
Line 90: Please consult a recent study related to this study by DiPietro et al. (2021):
Animals 2021, 11(10), 2859; https://doi.org/10.3390/ani11102859
Line 99: please define “same location”. Same barn, same county, etc.?
Line 102: because this is the first study of this nature, it is important to provide more details on factors influencing the microbiota to be used as comparison for future studies. For example, the time of the year, the type of pasture, any supplementation the foals were receiving, level of exercise, deworming, etc.
Line 103: how samples were shipped? How long between sampling and freezing?
Line 104: what do you mean with “samples were mixed”? Maybe homogenized would be better to avoid the impression of pooling.
Line 213: It is intuitive that the study is showing novelty, but why unexpected diversity? Are you actually showed higher diversity than previous studies?
Line 218: were the most abundant phyla found in this order?
Figure 1: the names in the phylogenetic tree are impossible to read.

·

Basic reporting

The manuscript by Gilroy et al. presents findings on the microbial community composition in horse gut/fecal microbiome. The authors evaluated the community compositions using shotgun metagenomics and assembled over 200 microbial and bacteriophage MAGs from five horse gut microbiome samples.
Overall, the manuscript is well written, and provides sufficient context for the work, knowledge gap being addressed, with suitable references where required. Language is clear for the reader to follow.
Raw data has been deposited in the SRA and accession numbers are provided.

The authors use the phrasing metagenomic species and MAGs interchangeably to reference the dereplicated metagenome assembled genomes. This makes the text a little difficult to follow at times, so I would recommend selecting one and using that throughout.
While the authors have extensively evaluated the taxonomic breakdown of the microbial community in the equine gut microbiome, the functional evaluation appears to be a bit preliminary and not comprehensive. The study could benefit from a more thorough assessment of the functional gene profiles of the microbes present in the gut microbiome to better understand the functional contribution of the microbes to equine health and fitness. It is also unclear what the motivation was to look into viral diversity and if there is apriori information on their importance in equine gut microbiomes, so additional background/references on this would be useful to include.

Line 242: The authors mention that the clustered MAGs span ten phyla. This is inconsistent with the referenced figure which shows just 8 phyla.
Line 245 can be rephrased.
Line 247: It is unclear to a reader who may not be familiar with GTDB what alphanumerical designations mean and/or its significance.
Line 264 & 265: Unclear which part of the figure represents this result.
Line 272: Missing citation for referenced study

Fig 1: The figure caption mentions “Hemicellulose substrates have been clustered and highlighted in red text” but there is no text highlighted in the figure, so it is unclear what is being referenced.
Fig 2a: Caption mentions abundance of MAGs, but the data displayed is relative abundance of MAGs in the community, so I would recommend rephrasing this.
Fig 2b: It is unclear how the species accumulation curve was constructed (methods are sparse)..
Fig 2c: Axis and legend can be more descriptive.
Supplementary Table 1 provides a summary of sequenced samples and the number of MAGs obtained from each. However, some of the numbers do not add up as a reader might expect e.g. columns R, S & T should add up to the total no. of bins (column Q). However, this is not the case since only bins above a certain threshold have been categorized. A better description of the columns/column headers would be helpful for the reader.
Supplementary Table 3: It is unclear what column Y represents. Descriptive column headers would be helpful.

Experimental design

The experiment is well designed to answer the questions of microbial community structure and function in equine gut microbiomes.
The authors have described the methods comprehensively to allow easy replication of the work.
The taxonomic evaluation performed here was based on just shotgun metagenomic sequencing and binning to retrieve metagenome assembled genomes (MAGs). A summary of how many total MAGs were retrieved from each sample, before dereplication, and what proportion of total reads assembled into MAGs is not included in the main text. This would be good information to provide to the reader and provides context on how comprehensively the sample was sequenced. (This is only available in the supplementary table – may consider moving Table S1 to main text.)
A few points where additional clarity may help:
- Fecal microbiota may be influenced by diet. It is unclear if all the horses sampled were fed the same kind (and which) diet. This can be clarified in the text.
- Unclear if any kind of host DNA removal was performed
- The authors may consider using an alternative reference database to run their Kraken taxonomic classification against (E.g. maxikraken2 https://lomanlab.github.io/mockcommunity/mc_databases.html). Since a large fraction of the reads remained unclassified using the RefSeq database, the use of a more comprehensive database comprised of sequences from non representative genomes and environmental isolates would help with improved classification of the raw reads.
- The authors compute and assign a quality score to the MAGs. May be good to describe the significance of that and why that is an additional criterion used, in the text.
- MAG level coverage is reported, however it would be useful to also report relative abundances of the MAGs in the different samples for sample-specific trends.
- Method used to compute species distribution and data used to generate Fig 2b is unclear.
- It is unclear which contigs were included in the the viral contig classification. This should be clarified, since it is not clear if the MAG contigs were excluded from this analysis.

Validity of the findings

Overall, I found the evaluation and description of the results from the MAGs to be not comprehensive.
This study and data generated provides ample opportunity to assess how many of the most/least abundant taxonomic lineages the authors were able to retrieve good quality genomes for, and if there were sample specific trends to what was captured in the bins. Additional evaluation of the lower quality bins could have provided additional hypotheses into why they the bins were prone to higher contamination (increased strain-level diversity?).
The results were heavy on assigning Candidatus names to the MAGs but did not go into much detail to describe trends in observed functional potential of the MAGs and how they could contribute to equine health/fitness.
In Line 281, a newly named class is described which does not contain representatives from vertebrate gut in NCBI. It would be very intriguing to see trends in relative abundance of this microbe in the different samples, and additional description of the functional potential of this microbe and how it compares to publicly available representatives.
The results of bacteriophage abundance in the samples is interesting. It would be useful to elucidate why they observed varying trends in phage abundance between the samples (E4 has the greatest number of phages, E5 has the least). What are the different sample specific factors that could contribute to this?
The authors acknowledge that a detailed evaluation has not been performed in this study, however there are several unaddressed points that would be useful to elaborate on, if the goal of the manuscript is to provide a detailed report of the taxonomic and more specifically functional diversity of equine gut microbiomes.

Additional comments

I would recommend additional assessment of the functional diversity of these MAGs to complete the story since the manuscript is currently heavy on just providing an overview of the taxonomic diversity and doesn’t delve much into functional potential.

Reviewer 3 ·

Basic reporting

Overall, this manuscript was very well-written and easy to follow. The language is nearly perfect with only minor grammatical errors and sometimes confusing syntax. I have listed these instances below.
Statements are well-cited and sufficient background information is provided. The article, figures, and tables are professionally structured and complete with raw data available. As no other literature, that I am aware of, regarding the equine gastrointestinal metagenome have yet been published, this article fills an important knowledge-gap. Though primarily descriptive, versus hypothesis-driven or mechanistic, given the scope of their work, I find this a valuable contribution to the field.

Please see suggestions for improvements below, as organized by line number:
- Line 59-60: “spread of human populations” - The point is understood but I believe this could be phrased differently to more clearly highlight your point. Perhaps “extension of human settlement”
- Line 64 - The jump from discussing equine therapy to slaughter is a bit jarring, I would recommend separating out these two ideas.
- Line 75 – The way this sentence is structured implies that the same number of human pathogens are just as likely to be harbored in the equine gut as equine pathogens, which is not the case. I would instead qualify human pathogens “equine and several human pathogens”
- Line 103 – Please define "bijous" for those not familiar
- Line 129 and 130 – Please define or describe SAM and BAM files for those not familiar
- Line 230 – Update tense, change show to showed
- Line 245 – Species is redundant in this line: ‘fourteen species of bacterial species’ – instead ‘fourteen of the bacterial species’?
- Line 263 – Missing end of parentheses
- Line 295 – Better characterize ‘each consecutive sample’ – unclear if this is multiple samples from one horse or each horse sample processed in succession
- Line 326 – Correction needed of "who’s" to "whose"


Figures:
- Figure 1a – Obviously challenging with so much information to include, but the text is very small in phylogenic tree. Is it possible to change layout so it is more legible? Also, it was unclear why the number of phyla included on the graphic (8) versus that stated in the text (10) were different - please address this discrepancy.
- Figure 2 – Please try to enlarge text, for all subsets of the figure, if possible.

- Supplemental:
- Line 74-76: The syntax and punctuation of this section is confusing, please reformat to make more clear.
- Section beginning Line 569/Description of Candidatus Darwinibacterium: missing closed end of parentheses
- Section beginning line 589 - also missing closing parentheses.
- Line 1052 – Italicized ‘this’ – please provide more explanation if intentional
- Section beginning line 1321 – phrasing and sentence structure is confusing for Candidatus Treponema, please reorganize to make more clear (similar issue to sections beginning line 569 and 589)
- Section beginning line 1331 – Similarly unclear syntax and appears misplaced period in line 1337.
- Section beginning line 1342 – Same issues as above. Please revise this section into complete and more clear sentences
- Section beginning line 1352 – Same issues as above re syntax and punctuation.

Experimental design

This paper fits well within the Aims and Scope of the journal. The exploratory research questions is well-defined, relevant, and as the first assessment of its kind, meaningful. This application of metagenomics to the equine fecal microbiome definitely fills a knowledge gap. The metagenomic methods applied are rigorous and sufficient detail is provided such that these steps are reproducible.

Validity of the findings

All underlying data have been provided. A larger sample size, as always, would have added significance to the findings but the cost of metagenomics makes this limitation understandable.
The conclusions are well-stated and generally supported though I do find them slightly over-reaching in one area (see comment below). Otherwise, findings are presented very well.

Suggested Improvement:
- Line 358 – I believe this statement is a bit over-reaching. Instead I would update to "... preliminary census of the horse gut microbiome specific to yearling thoroughbreds"

Additional comments

I enjoyed reading this manuscript. While I was not surprised to read that the equine fecal microbiome is very diverse and home to many yet-to-be-characterized species, this group's demonstration of these facts using advanced sequencing is a very useful addition to the equine gut microbiome research field. The writing is clear with only minor grammatical and syntax suggestions. The figures are understandable and applicable, though are a bit difficult to read in their current 'print' format and thus could benefit, as much as possible, from slight adjustments to text size. Thank you for the opportunity to review this manuscript and please let me know if there is any further information I can provide.

---

## Round 0.2 · Minor Revisions

Please address the queries raised by Reviewer 2.

·

Basic reporting

All my questions have been answered.

Experimental design

The experimental design was not changed.

Validity of the findings

The same as previous.

Additional comments

No comments.

·

Basic reporting

No comment

Experimental design

No comment

Validity of the findings

No comment

Additional comments

The authors have addressed the comments provided in the first round of review. One of the biggest points raised was that the manuscript did not provide a comprehensive assessment of the functional potential of the equine gut microbiome. The authors have clarified this and the manuscript has been restructured to focus on a comprehensive description of the taxonomic diversity. This makes the manuscript more focused and sets up the study as a resource for future exploration in this field.
I have one additional clarification - relative abundances of each of the obtained MAGs in the respective samples are provided in Table S7. Relative abundances add up to values > 1. This seems incorrect and should probably be double checked and/or clarified and results in line 497 corrected accordingly.

Reviewer 3 ·

Basic reporting

I enjoyed reading this revised manuscript and believe that the suggested revisions have been well-integrated. The writing is now perfect apart from one missed comma (see below) and very easy to follow. References and background information have been enhanced for this revision as have the figures, tables, and general structure of the paper. The raw data remains accessible and the results are relevant to the exploratory hypothesis.

Suggested change:
Line 90: Suggest changing "vast only superficially" to "vast, only superficially"

Experimental design

This paper continues to fit well within the Aims and Scope of the journal. The exploratory research questions is well-defined, relevant, and, as the first assessment of its kind, meaningful. The metagenomic methods have been enhanced, in response to other reviewers' comments and, with additional detail regarding methodology added to this version, the work is reproducible.

Validity of the findings

These findings, and their processing, are valid and accessible to those who may wish to apply their own evaluations.

Additional comments

I thank the authors for their revisions and congratulate them on the work they have done. No further comments.

---

## Round 0.3 · accepted · Accept

I feel happy to recommend the manuscript for publication in PeerJ.